# Regulation of substrate utilization and adiposity by Agrp neurons

João Paulo Cavalcanti-de-Albuquerque[1,2], Jeremy Bober[1], Marcelo R. Zimmer[1,3] & Marcelo O. Dietrich[1,3,4]

The type of nutrient utilized by the organism at any given time—substrate utilization—is a critical component of energy metabolism. The neuronal mechanisms involved in the regulation of substrate utilization in mammals are largely unknown. Here, we found that activation of hypothalamic Agrp neurons rapidly altered whole-body substrate utilization, increasing carbohydrate utilization, while decreasing fat utilization. These metabolic changes occurred even in the absence of caloric ingestion and were coupled to increased lipogenesis. Accordingly, inhibition of fatty acid synthase—a key enzyme that mediates lipogenesis— blunted the effects of Agrp neuron activation on substrate utilization. In pair-fed conditions during positive energy balance, activation of Agrp neurons improved metabolic efficiency, and increased weight gain and adiposity. Conversely, ablation of Agrp neurons impaired fat mass accumulation. These results suggest Agrp neurons regulate substrate utilization, contributing to lipogenesis and fat mass accumulation during positive energy balance.

[1] Department of Comparative Medicine, Yale University School of Medicine, 310 Cedar Street, Brady Memorial Laboratory Room 410, New Haven, CT 06520, USA. [2] Institute of Biophysics Carlos Chagas Filho and of Nutrition Josue de Castro, Universidade Federal do Rio de Janeiro, Rio de Janeiro, RJ 21941, Brazil. [3] Graduate Program in Biochemistry, Universidade Federal do Rio Grande do Sul, Porto Alegre, RS 90035, Brazil. [4] Department of Neuroscience, Yale University School of Medicine, 333 Cedar Street, New Haven, CT 06520, USA. Correspondence and requests for materials should be addressed to M.O.D. (email: marcelo.dietrich@yale.edu)

O besity is a major health problem that results from altered regulation of energy balance. Energy balance is the relation between energy intake and energy expenditure. When energy balance is positive (intake>expenditure), there is an increase in energy storage and fat accumulation. A chronic state of positive energy balance leads to obesity. However, a less appreciated component in the regulation of energy balance is the selection of the type of substrate—typically carbohydrate versus fat—used in metabolic reactions. This substrate selection (or substrate utilization) is important as different energy substrates yield different amounts of free energy. Impairment in the capacity to shift between different energy substrates is linked to obesity[1]. Thus, it is important to identify how the organism controls substrate utilization for a better understanding of the regulation of energy balance and obesity development.

Energy metabolism is tightly controlled by the hypothalamus. Agouti-related peptide producing neurons (hereafter, Agrp neurons)[2–5] in the hypothalamus regulate energy metabolism by responding to a variety of circulating factors[6–9]. These neurons are active during food deprivation[2,10–12] and are strongly associated to control of food intake[2,5,13–18]. However, recent work suggested that Agrp neurons also regulate other metabolic processes, including white adipose tissue (WAT) browning[19] and brown adipose tissue glucose metabolism[20] and thermogenesis[19,21]. During our previous studies investigating the function of Agrp neurons[14,19,22], we observed rapid changes in whole-body substrate utilization upon activation of Agrp neurons. Based on these unexpected observations, we investigated the involvement of Agrp neurons in the regulation of peripheral substrate utilization and lipogenesis.

Here, we report that Agrp neurons rapidly shift whole-body metabolism towards lipid storage, a mechanism we suggest as being important for fat accumulation during positive energy balance (i.e., a metabolic state coupled to weight gain).

## Results

### Acute switch in nutrient utilization upon Agrp neuron activation.

To gain insight into the acute regulation of metabolism by Agrp neurons, nutrient utilization was measured by indirect calorimetry upon activation of these neurons. $Agrp^{Trpv1}$ mice (as previously characterized[14,19]) were used to specifically activate Agrp neurons by selectively expressing the capsaicin-sensitive channel, Trpv1, which is transiently activated by peripheral injection of capsaicin. Activation of Agrp neurons by capsaicin rapidly increased food intake in $Agrp^{Trpv1}$ mice (Fig. 1a and Supplementary Figure 1)[14]. We also observed a sharp increase in respiratory exchange ratio (RER)—calculated as the ratio (VCO2/VO2)—in these animals (Fig. 1b), in line with the observed fast kinetics of feeding, suggesting an acute shift in substrate utilization towards carbohydrates relative to fat. Based on gaseous exchange[23], we calculated total rates of fat utilization and carbohydrate utilization for the whole animal. In line with changes in RER, activation of Agrp neurons led to a rapid and prolonged decrease in fat utilization (Fig. 1c) and increase in carbohydrate utilization (Fig. 1d). Because this experiment was performed in the presence of food and the effects on nutrient utilization were prolonged compared to feeding upon Agrp neuron activation (Fig. 1a)[14], these metabolic shifts could be due to changes in postprandial metabolism. Using linear regression analysis between the initial bout of food intake (0–30 min) and the subsequent RER (30–60 min), we could not find a positive correlation between these parameters (Fig. 1e). However, the lack of positive correlation could be explained by a ceiling effect.

We did not observe statistically significant changes in $VO_2$ (Fig. 1f), $VCO_2$ (Fig. 1g) or energy expenditure (Fig. 1h). However, non-statistical decreases in $VO_2$ and increases in $VCO_2$ accounted for the changes in RER (inserts Fig. 1f, g). We also measured activity levels upon Agrp neuron activation[14] and did not observe statistical changes in ambulatory activity (Fig. 1i). Thus, these results suggest Agrp neurons rapidly control whole-body substrate utilization, an effect that could be the consequence of postprandial metabolic shifts.

### Agrp neuron activation control substrate utilization in the absence of ingestion.

We further explored the extent to which activation of Agrp neurons and carbohydrate ingestion interact to rapidly shift metabolism (Supplementary Figure 2 and Supplementary Note 1). We infused both control and $Agrp^{Trpv1}$ mice with glucose ($2\,g\,kg^{-1}$ body weight, via gavage) and activated Agrp neurons by injecting capsaicin (Fig. 2a). Food was removed from the cage during the experiment to prevent ingestion. Activation of Agrp neurons led to an increased peak and a more sustained elevation of RER (Fig. 2b, c), an effect that was present even in mice that did not receive glucose infusion (Fig. 2b, c). Accordingly, activation of Agrp neurons led to a sustained decrease in fat utilization (Fig. 2d, e) and increase in carbohydrate utilization (Fig. 2f, g), regardless of glucose infusion. These results suggest that activation of Agrp neurons alone is capable of promoting shifts in substrate utilization.

In the previous experiments, even in animals infused with saline via gavage, small amounts of liquid were delivered, raising the possibility that acute gastric distension acts together with Agrp neuron activation to promote changes in substrate utilization. To exclude this possibility, we repeated our experiments in a new cohort of mice in which Agrp neurons were activated in the absence of food (Fig. 3a). In line with our previous findings, activation of Agrp neurons induced an increase in RER (Fig. 3b), a decrease in fat utilization (Fig. 3c) and an increase in carbohydrate utilization (Fig. 3d). In support of these results, activation of Agrp neurons using DREADD (designer receptor exclusively activated by designer drugs) also showed similar shifts in substrate utilization (Supplementary Figure 3 and Supplementary Note 2).

We measured ambulatory activity and found an increase upon activation of Agrp neurons in the absence of food (Fig. 3e), suggestive of foraging behaviors[14,16,24]. Importantly, activity levels between control and $Agrp^{Trpv1}$ mice started to diverge around 20–25 min after capsaicin injection, while RER started to diverge around 10–15 min. The fact that RER started to diverge before activity levels suggests that the effect of Agrp neuron activation on RER is independent of an increase in locomotion. To further rule out the effects of activity on RER, we used linear regression analysis (Fig. 3f, g). We found that activity levels and RER were weakly correlated in both control and $Agrp^{Trpv1}$ mice (control: $r^2 = 0.244$, $F_{1,63} = 20.12$, $P < 10^{-4}$; $Agrp^{Trpv1}$: $r^2 = 0.155$, $F_{1,63} = 11.21$, $P < 10^{-3}$). However, the slopes were not statistically different from each other (control: slope $= 1.68 \times 10^{-4} \pm 0.37 \times 10^{-4}$; $Agrp^{Trpv1}$: slope $= 1.21 \times 10^{-4} \pm 0.36 \times 10^{-4}$; $F_{1,126} = 0.82$, $P = 0.36$), suggesting the physiological relationship between RER and activity levels is not altered by Agrp neuron activation. Importantly, across different activity levels, activation of Agrp neurons increased RER, as evidenced by the parallel linear regression lines and by statistically different intercepts (control: intercept $= 0.766 \pm 0.005$; $Agrp^{Trpv1}$: intercept $= 0.878 \pm 0.007$, $F_{1,127} = 201.5$, $P < 10^{-4}$) (Fig. 3f, g). Thus, these results demonstrate that activation of Agrp neurons increases RER

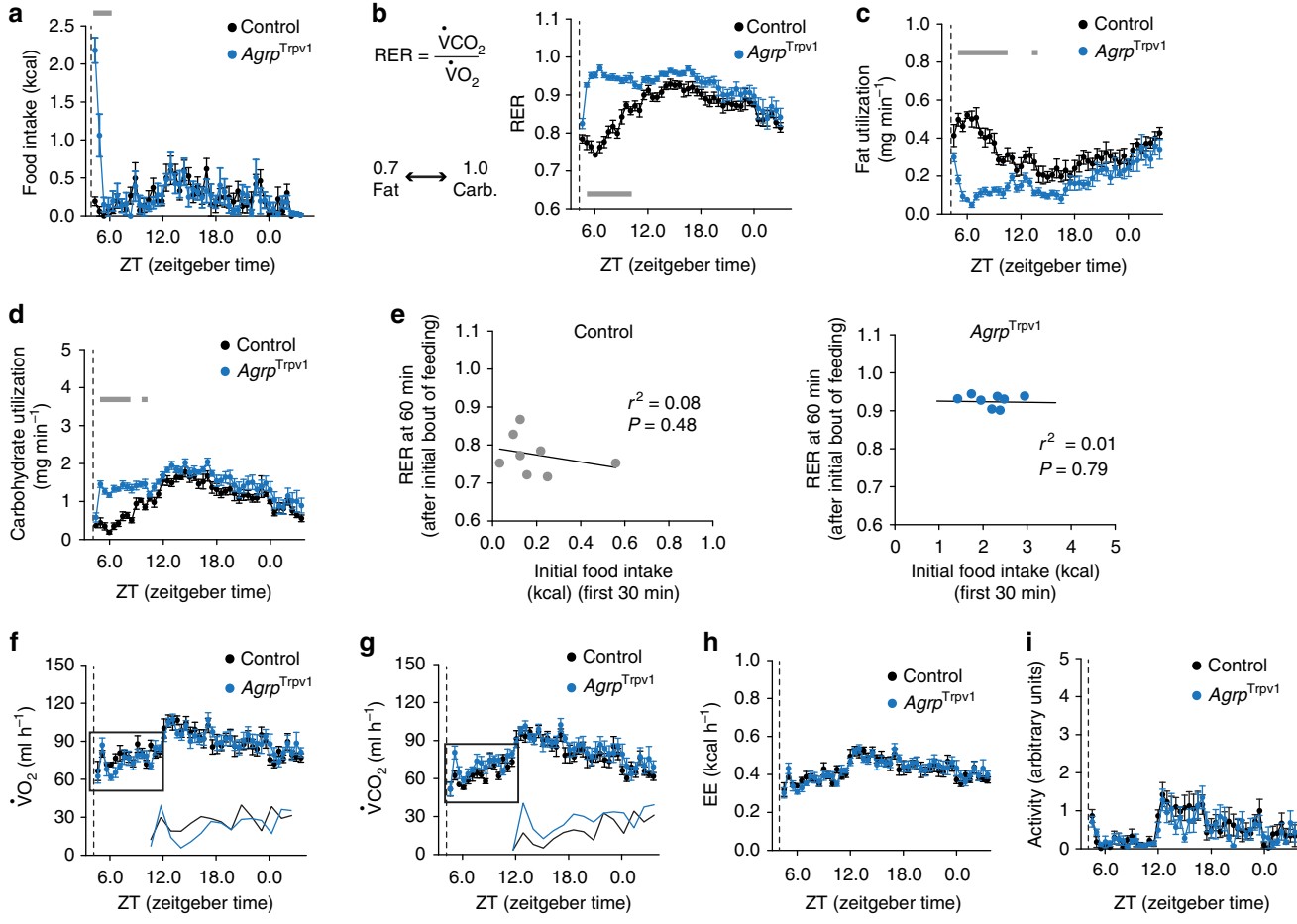

**Fig. 1** Rapid shift in substrate utilization upon activation of agouti-related peptide (Agrp) neurons. **a** Food intake. **b** Respiratory exchange ratio (RER) (interaction: $F_{45, 630} = 8.40$, $P < 0.0001$; time: $F_{45, 630} = 13.86$, $P < 0.0001$; group: $F_{1, 14} = 52.59$, $P < 0.0001$). **c** Calculated fat utilization (interaction: $F_{45, 630} = 6.44$, $P < 0.001$; time: $F_{45, 630} = 7.22$, $P < 0.0001$; group: $F_{1, 14} = 33.83$, $P < 0.0001$). **d** Calculated carbohydrate utilization (interaction: $F_{45, 630} = 2.77$, $P < 0.0001$; time: $F_{45, 630} = 18.48$, $P < 0.0001$; group: $F_{1, 14} = 16.73$, $P = 0.001$). **e** Linear regression analysis between the size of the initial bout of food intake (first 30 min) and RER (at the 60 min measurement) in control and $Agrp^{Trpv1}$ mice. **f** $VO_2$ (interaction: $F_{45, 630} = 1.07$, $P = 0.34$; time: $F_{45, 630} = 11.22$, $P < 0.0001$; group: $F_{1, 4} = 0.01$, $P = 0.89$). **g** $VCO_2$ (interaction: $F_{45, 630} = 0.88$, $P = 0.68$; time: $F_{45, 630} = 15.07$, $P < 0.0001$; group: $F_{1, 14} = 1.58$, $P = 0.22$). In **f** and **g**, inserts show the mean $VO_2$ and $VCO_2$. **h** Energy expenditure (interaction: $F_{45, 630} = 0.96$, $P = 0.53$; time: $F_{45, 630} = 12.09$, $P < 0.0001$; group: $F_{1, 14} = 0.02$, $P = 0.87$). **i** Ambulatory activity (interaction: $F_{45, 630} = 0.80$, $P = 0.81$; time: $F_{45, 630} = 6.02$, $P < 0.0001$; group: $F_{1, 14} = 0.31$, $P = 0.58$). Statistical analysis was performed using two-way analysis of variance (ANOVA) with time as a repeated measure followed by Holm–Sidak's multiple comparisons test (MCT). Gray bars indicate time points in which MCTs were statistically significant ($P < 0.05$). From **a**–**i** control (black; $n = 8$) and $Agrp^{Trpv1}$ mice (blue; $n = 8$). Dashed line indicates time of capsaicin injection. Symbols indicate mean ± SEM

independently of the ingestion of food, an effect that cannot be fully explained by changes in activity levels.

**Participation of lipogenesis in Agrp neuron-mediated shifts in metabolism**. We next explored biochemical changes in peripheral tissues that could be mechanistically linked to the observed shifts in metabolism upon activation of Agrp neurons. We found that in the absence of food ingestion, Agrp neuron activation decreased circulating levels of non-esterified fatty acids (NEFAs, Fig. 4a, b) with no changes in basal blood glucose levels (Fig. 4c). The lack of changes in basal blood glucose levels do not exclude the possibility that Agrp neurons control glucose handling during glucose challenge[20]. Because circulating NEFAs decrease upon Agrp neuron activation, these results suggest a decrease in release, and possibly an increase in deposition of fat. To test this hypothesis, we measured expression levels of genes involved in lipid metabolism in the WAT from $Agrp^{Trpv1}$ and control mice 60 min after capsaicin injection. We found a decrease in the expression level of *Ppara* (Fig. 4d), a gene involved in the promotion of fat

catabolism. We also found a significant increase in expression levels of *hexokinase2* (*hk2*; Fig. 4d), a rate-limiting enzyme involved in glycolysis, a critical metabolic step to provide carbons for de novo lipogenesis. Hormone-sensitive lipase (HSL) is an essential step in the breakdown of triglycerides to release fatty acids in circulation. Activation of hormone-sensitive lipase occurs by phosphorylation of this enzyme in several serine residues. Upon activation of Agrp neurons, we found decreased levels of phosphorylated hormone-sensitive lipase (p-HSL) in WAT compartments (Fig. 4e and Supplementary Figure 6). Together, these experiments indicate activation of Agrp neurons leads to increased lipogenesis and decreased lipolysis in the WAT.

De novo lipogenesis can drive RER above 1.0[23], and could be a potential factor involved in Agrp neuron activation-mediated shifts in RER. To test for the participation of fat synthesis in the rapid effects of Agrp neurons on metabolism, we blocked fatty acid synthase (FAS), a key enzyme involved in fat storage[25]. We treated mice with a pharmacological inhibitor of fatty acid synthase (C75, 10 mg kg⁻¹, i.p.)[26,27] and activated Agrp neurons

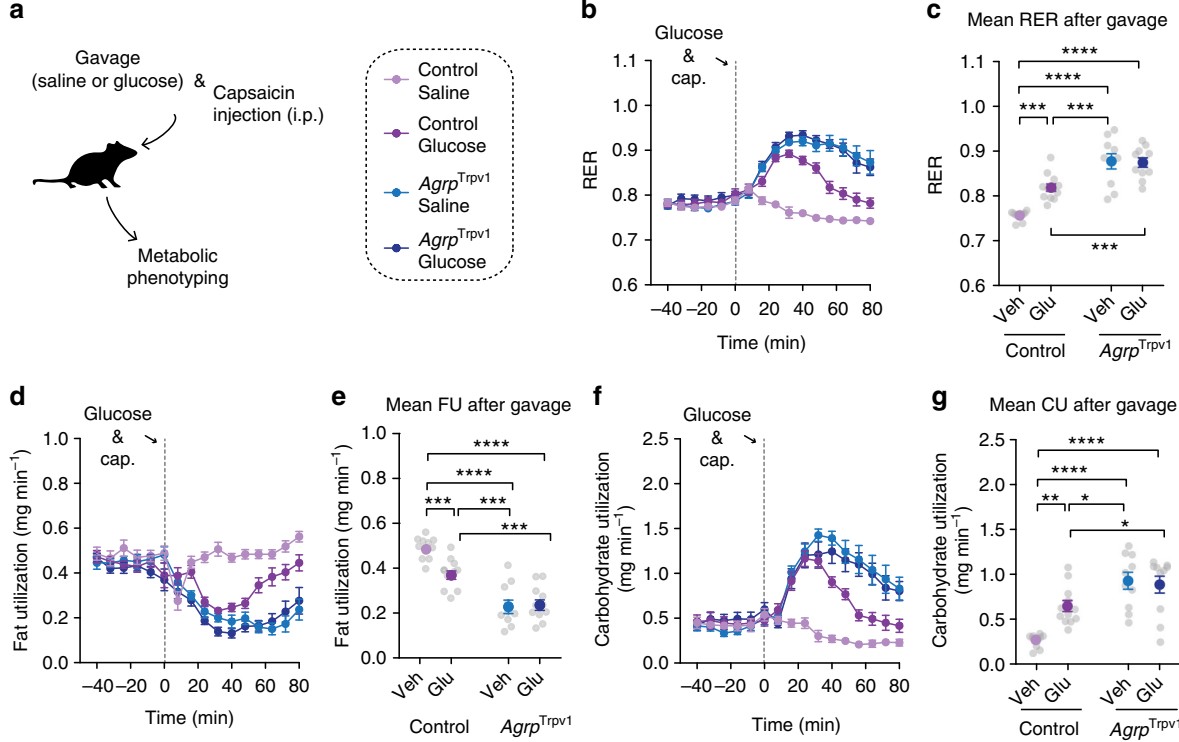

**Fig. 2** Glucose ingestion and Agrp neurons control substrate utilization independently. **a** Control and Agrp[Trpv1] mice received a bolus of saline or glucose (2 g kg[−1]) via gavage followed by peripheral injection of capsaicin (10 mg kg[−1], intraperitoneal (i.p.)). **b** Respiratory exchange ratio (RER). **c** Mean RER after gavage (interaction: $F_{1, 40} = 9.82$, $P = 0.003$; gavage solution: $F_{1, 40} = 8.20$, $P = 0.006$; genotype: $F_{1, 40} = 71.31$, $P < 0.0001$). **d** Fat utilization. **e** Mean fat utilization after gavage (interaction: $F_{1, 40} = 7.91$, $P = 0.007$; gavage solution: $F_{1, 40} = 6.07$, $P = 0.01$; genotype: $F_{1, 40} = 79.82$, $P < 0.0001$). **f** Carbohydrate utilization. **g** Mean carbohydrate utilization after gavage (interaction: $F_{1, 40} = 8.29$, $P = 0.006$; gavage solution: $F_{1 40} = 5.26$, $P = 0.02$; genotype: $F_{1, 40} = 37.44$, $P < 0.0001$). In **c**, **e** and **g** statistical analysis was performed using two-way analysis of variance (ANOVA) on the mean response after gavage and capsaicin injection; genotype (control vs. Agrp[Trpv1]) and gavage infusion (saline vs. glucose) were used as factors for the ANOVA. Holm–Sidak's multiple comparisons test (MCT) was used to find post hoc differences among groups. MCTs are indicated as *$P < 0.05$, **$P < 0.01$, ***$P < 0.001$ and ****$P < 0.0001$ in figure panels. Control mice + saline gavage ($n = 11$); control mice + glucose gavage ($n = 12$); Agrp[Trpv1] mice + saline gavage ($n = 10$); Agrp[Trpv1] mice + glucose gavage ($n = 11$). Dashed gray line indicates time of oral gavage and capsaicin injection

in indirect calorimetry chambers (Fig. 4f). Treatment of control mice with the fatty acid synthase inhibitor had no effects on RER (Fig. 4g), fat utilization (Fig. 4j) or carbohydrate utilization (Fig. 4m). The lack of effects of fatty acid synthase inhibition on substrate utilization is in line with the low levels of lipogenesis during the light cycle of mice. In contrast to control animals, inhibition of fatty acid synthase blocked the effects of Agrp neuron activation on substrate utilization (Fig. 4h, i, k, l, n, o). These results provide further support for the rapid shift in metabolism towards lipogenesis mediated by Agrp neuron activation in mice.

**Sympathetic signaling mediates peripheral effects of Agrp neurons**. Norepinephrine release and binding to adrenergic receptors on fat compartments promotes lipolysis, while its inhibition favors lipogenesis. Accordingly, we predicted that Agrp neurons control substrate utilization and lipogenesis by inhibiting sympathetic signaling on fat compartments in anabolic states. Here, we treated mice with a β3-adrenergic receptor agonist (CL 316,243)[28] (Fig. 5a), which are highly selective to fat compartments[29], while activating Agrp neurons (Fig. 5a). Treatment of control mice with CL 316,243 did not alter RER (Figs. 5b and 5d) but prevented the increase in RER upon Agrp neuron activation in Agrp[Trpv1] mice (Fig. 5c-d). When we calculated fat utilization, CL 316,243 completely reverted the inhibition of whole-body fat utilization upon Agrp neuron activation (Fig. 5e, g) but had no effects in control animals (Fig. 5f, g). Concomitantly, CL 316,243

prevented the increase in carbohydrate utilization upon activation of Agrp neurons (Fig. 5h–j). Thus, promotion of β3-adrenergic receptor signaling using a pharmacological agonist reverts the effects of Agrp neuron activation on peripheral fuel metabolism.

Next, we tested whether CL 316,243 could acutely block the effects of Agrp neuron activation on food intake. We first performed a dose response study to detect the range of CL 316,243 doses that could revert the increase in RER upon Agrp neuron activation. We found that doses as low as 0.01 mg kg[−1] almost completely reverted the effects of Agrp neurons on RER (Fig. 5k). Next, we selected two doses of CL 316,243 (1.00 and 0.01 mg kg[−1]) to investigate its effects on food intake mediated by Agrp neuron activation. In all conditions tested, CL 316,243 did not revert the effects of Agrp neuron activation on food intake (Fig. 5l, m). Of note, CL 316,243 is highly anorexigenic[29], but its effects are observed after several hours and not as rapid as the effects of Agrp neuron activation on feeding[14–16]. These findings suggest a divergence between the feeding and metabolic mechanisms underlying Agrp neuron function and support the argument that Agrp neuron activation favors the storage of fat (lipogenesis) in situations of energy surfeit by suppressing sympathetic activity.

**Agrp neuron activation potentiate adiposity in obesogenic conditions**. Since Agrp neuron activity shifts metabolism toward lipogenesis, it is expected that during prolonged neuronal activation mice will accumulate more fat than control counterparts.

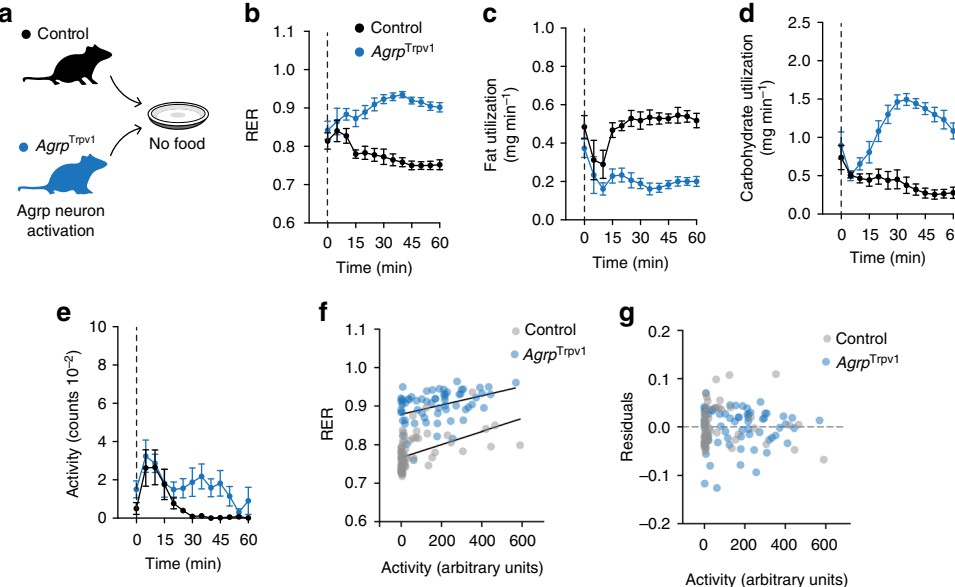

**Fig. 3** Agrp neuron activation induces a shift in substrate utilization. **a** Control (black, $n = 5$) and $Agrp^{Trpv1}$ (blue, $n = 5$) mice were tested in indirect calorimetry chambers upon injection of capsaicin without food provided. **b** Respiratory exchange ratio (RER) (interaction: $F_{12, 96} = 8.96$, $P < 0.0001$; time: $F_{12, 96} = 1.31$, $P = 0.22$; genotype: $F_{1, 8} = 54.45$, $P < 0.0001$). **c** Calculated fat utilization (interaction: $F_{12, 96} = 3.52$, $P = 0.0002$; time: $F_{12, 96} = 3.23$, $P = 0.0006$; genotype: $F_{1, 8} = 55.31$, $P < 0.0001$). **d** Calculated carbohydrate utilization (interaction: $F_{12, 96} = 13.52$, $P < 0.0001$; time: $F_{12, 96} = 6.25$, $P < 0.0001$; genotype: $F_{1, 8} = 58.16$, $P < 0.0001$). **e** Ambulatory activity (interaction: $F_{12, 96} = 1.12$, $P = 0.34$; time: $F_{12, 96} = 7.13$, $P < 0.0001$; genotype: $F_{1, 8} = 5.29$, $P = 0.05$). **f** Linear regression models of activity levels (in arbitrary units) and RER in control (gray) and $Agrp^{Trpv1}$ (red) mice. **g** Residuals from the linear regression model. In **b**, **c**, **d** and **e**, statistical analysis was performed using two-way analysis of variance (ANOVA) with time as a repeated measure followed by Holm–Sidak's multiple comparisons test (MCT; not shown). Dashed line indicates time of capsaicin injection. Symbols indicate mean ± SEM

To test this assumption, we daily activated Agrp neurons in $Agrp^{Trpv1}$ mice and measured body weight and fat mass changes over 10 days in ad libitum fed animals. Repeated activation of Agrp neurons led to an increase in food intake, body weight gain, fat mass and metabolic efficiency (Supplementary Figure 4), i.e., the amount of body weight gained related to the amount of energy ingested.

To further dissect the discrete contribution of Agrp neurons to adiposity and weight gain during obesogenic conditions, we set out to selectively activate Agrp neurons under controlled positive energy balance. We took advantage of the physiological hyperphagia that occurs when rodents are switched from a normal chow diet to a high-fat diet in the laboratory[30,31]. In our colony, this acute dietary switch led to an increase of ~86% in the amount of overnight calories consumed (Fig. 6a). Based on these data, we designed a pair-fed study in which control and $Agrp^{Trpv1}$ mice could be fed more calories than they would usually consume under a regular chow diet, thus inducing a state of positive energy balance (Fig. 6b). When control and $Agrp^{Trpv1}$ mice were fed an isocaloric high-fat diet (compared to their ad libitum normal chow intake), activation of Agrp neurons did not lead to changes in body weight gain or metabolic efficiency (Fig. 6c). However, when pair-fed 25% or 50% more calories than their ad libitum food intake under a normal chow diet, activation of Agrp neurons in $Agrp^{Trpv1}$ mice increased body weight gain and metabolic efficiency compared to littermate controls (Fig. 6d, e).

When switched to a high-fat diet, mice displayed hyperphagia lasting 5 days (Fig. 6f)[30,31]. We took advantage of this phenomenon to study mice in positive energy balance in a more prolonged pair-fed experiment (Fig. 6g). Using this approach, we controlled the number of calories ingested by control and Agrp neuron activated mice during 5 days of hyperphagia (Fig. 6h). Despite identical food intake, activation of Agrp neurons increased body weight gain compared to control mice (Fig. 6i).

We then measured the metabolic efficiency and changes in fat mass at the end of the study (day 6) compared to day 0—when we introduced the high-fat diet. In this experimental setting, activation of Agrp neurons increased metabolic efficiency (Fig. 6j) and fat mass gain (Fig. 6k) with no changes in lean mass (delta lean mass: control = $0.26 \pm 0.39$, $n = 8$; $Agrp^{Trpv1} = 0.24 \pm 0.20$, $n = 8$; $t_{14} = 0.04$, $P = 0.96$, t-test). In support of these findings, we found similar results in $Agrp^{hM3Dq}$ mice (Supplementary Figure 5 and Supplementary Note 3).

**Ablation of Agrp neurons impairs adiposity.** Finally, we tested whether the loss of function of Agrp neurons would impair adiposity in conditions of positive energy balance. We studied control and $Agrp^{DTR}$ mice[17], allowing controlled ablation of these neurons. In adults, ablation of Agrp neurons leads to aphagia and death[17]. To bypass the aphagia after Agrp neuron ablation, we devised an enteral feeding scheme using a gastrostomy tube to deliver controlled amounts of liquid diet into the stomach of mice (Fig. 7a). After tube implantation, mice were allowed to recover from surgery and acclimate to liquid diet for 2 days. We then started to gradually increase the amount of diet infused in the stomach, until reaching positive energy balance (animals started to gain body weight). At post-surgery days 7 and 9, we injected diphtheria toxin ($50 \mu g\, kg^{-1}$, intramuscular) to ablate Agrp neurons. In this feeding regimen, ablation of Agrp neurons did not lead to death and mice continued to increase body weight (Fig. 7b). However, ablation of Agrp neurons led to a significant decrease in fat mass gain (Fig. 7c) but not lean mass (delta lean mass: control = $-0.07 \pm 0.30$, $Agrp^{DTR} = 0.13 \pm 0.49$; $U = 9$, $P = 0.45$, Mann–Whitney test) as measured by repeated magnetic resonance imaging (MRI) scans before the first diphtheria toxin injection and at the end of the study. At 6 days after the first injection of diphtheria toxin, we dissected the fat tissue for

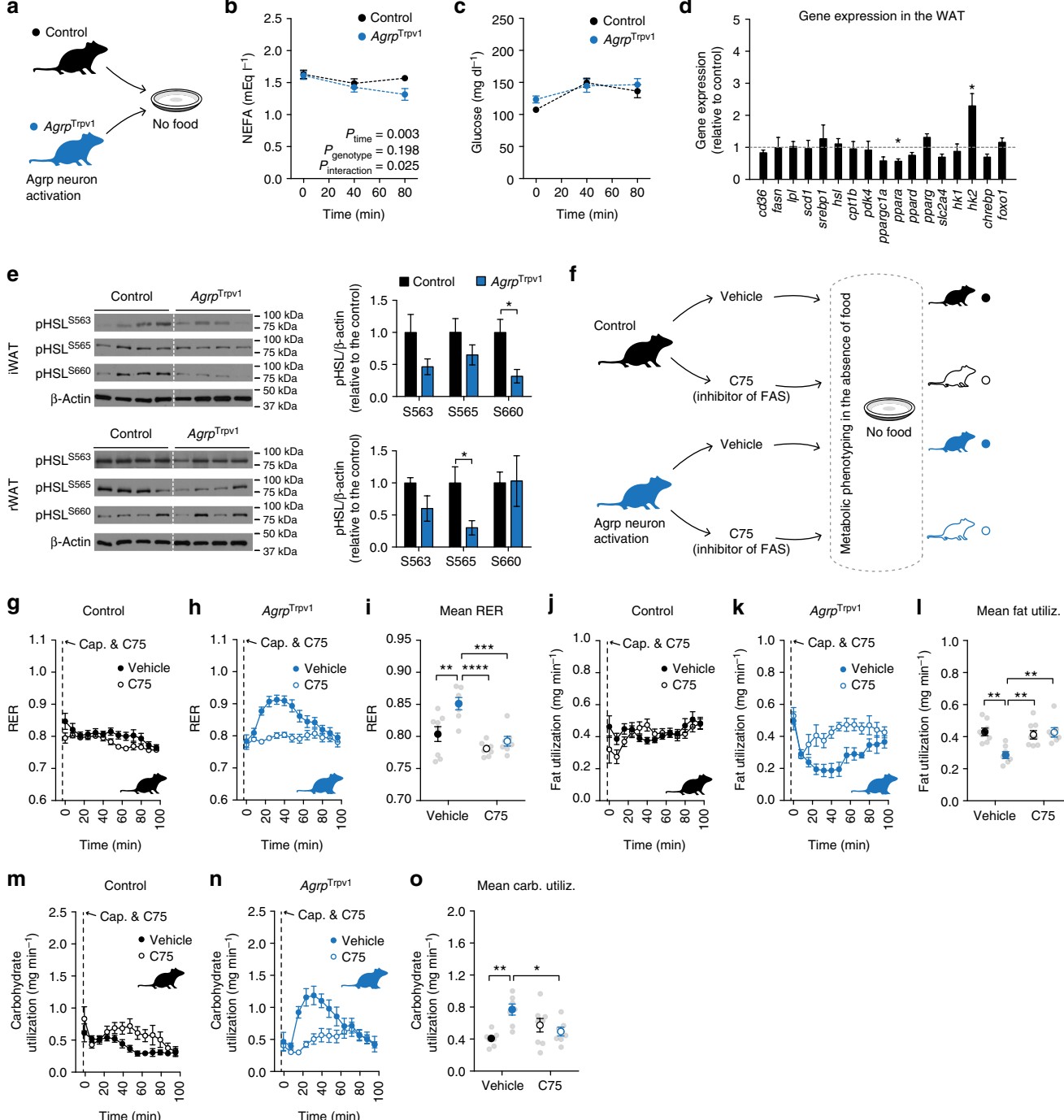

**Fig. 4** Agrp neurons promote lipogenesis. **a** Control (black) and $Agrp^{Trpv1}$ (blue) mice were injected with capsaicin. **b** NEFA levels (interaction: $F_{2, 54} = 3.91$, $P = 0.02$; time: $F_{2, 54} = 9.64$, $P = 0.0003$; genotype: $F_{1, 27} = 1.74$, $P = 0.19$; MCT: $P_{80 min} = 0.03$). **c** Blood glucose levels (interaction: $F_{2, 54} = 1.47$, $P = 0.23$; time: $F_{2, 54} = 14.6$, $P < 0.0001$; genotype: $F_{1, 27} = 0.67$, $P = 0.41$; MCT: ns). In **b** and **c**: control ($n = 15$) and $Agrp^{Trpv1}$ mice ($n = 14$). Statistics: two-way analysis of variance (ANOVA) with time as a repeated measure followed by Holm–Sidak's multiple comparisons test (MCT). **d** Gene expression in the WAT. Data are normalized to control levels (dashed line). Statistics: Student's t-test. *$P < 0.05$. **e** Western blotting analysis of phosphorylated HSL in inguinal WAT and retroperitoneal WAT ($n = 4$ mice per group). Blots were cropped for clarity. Statistics: Student's t-test. *$P < 0.05$. **f** Control and $Agrp^{Trpv1}$ mice randomly received vehicle or the FAS inhibitor (C75, 10 mg kg$^{-1}$, intraperitoneal (i.p.)) immediately before capsaicin injection. **g–i** Respiratory exchange ratio (RER) (interaction: $F_{1, 26} = 4.36$, $P = 0.04$; drug: $F_{1, 26} = 22.28$, $P < 0.0001$; genotype: $F_{1, 26} = 11.73$, $P = 0.002$). **j–l** Fat utilization (interaction: $F_{1, 26} = 9.97$, $P = 0.04$; drug: $F_{1, 26} = 6.00$, $P = 0.02$; genotype: $F_{1, 26} = 6.56$, $P = 0.01$). **m–o** Carbohydrate utilization: (interaction: $F_{1, 26} = 12.18$, $P = 0.001$; drug: $F_{1, 26} = 0.76$, $P = 0.38$; genotype: $F_{1, 26} = 4.90$, $P = 0.03$). In **g–o**, statistics: two-way ANOVA on the mean response after drug and capsaicin injection; genotype and drug as factors for the ANOVA. Holm–Sidak's multiple comparisons test (MCT) was used and indicated as *$P < 0.05$, **$P < 0.01$, ***$P < 0.001$ and ****$P < 0.0001$. Control mice + vehicle ($n = 8$); control mice + C75 ($n = 8$); $Agrp^{Trpv1}$ mice + vehicle ($n = 7$); $Agrp^{Trpv1}$ mice + C75 ($n = 7$). Dashed line indicates time of injections. Colored symbols indicate mean ± SEM. Gray symbols indicate individual values

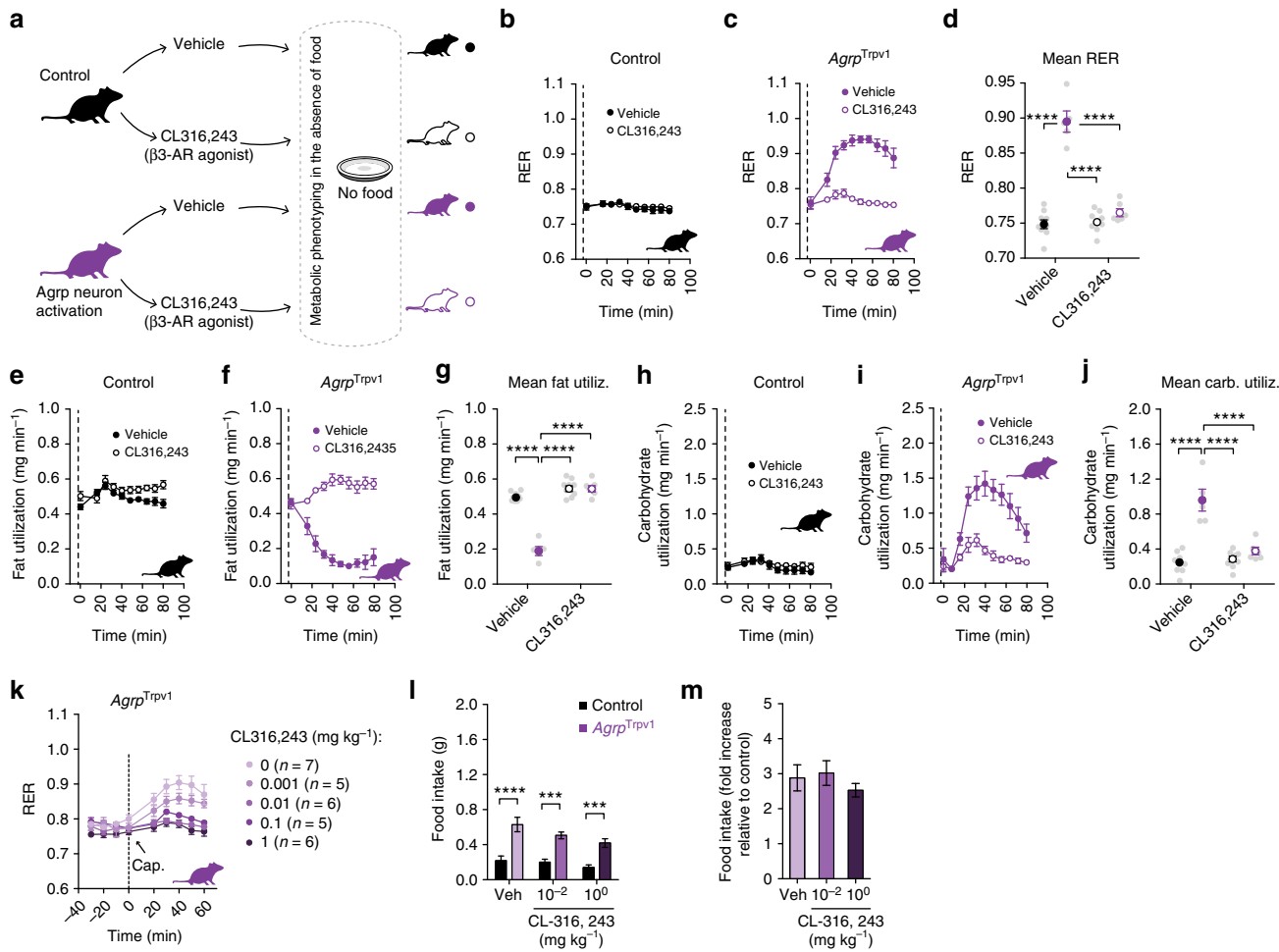

**Fig. 5** Sympathetic signaling is involved in peripheral effects of Agrp neurons. **a** Control (black) and Agrp^Trpv1 mice (purple) were randomized to receive vehicle or the β3-adrenergic receptor agonist (CL 316,243, 1 mg kg$^{-1}$, intraperitoneal (i.p.)); vehicle or CL 316,243 were injected immediately before capsaicin. **b–d** Respiratory exchange ratio (RER) (interaction: $F_{1, 25} = 75.09$, $P < 0.0001$; drug: $F_{1, 25} = 67.96$, $P < 0.0001$; genotype: $F_{1, 25} = 108.4$, $P < 0.0001$). **e–g** Fat utilization (interaction: $F_{1, 25} = 85.56$, $P < 0.0001$; drug: $F_{1, 25} = 150.4$, $P < 0.0001$; genotype: $F_{1, 25} = 86.42$, $P < 0.0001$). **h–j** Carbohydrate utilization (interaction: $F_{1, 25} = 29.74$, $P < 0.0001$; drug: $F_{1, 25} = 22.75$, $P < 0.0001$; genotype: $F_{1, 25} = 50.12$, $P < 0.0001$). In **d**, **g** and **j**, statistics: two-way analysis of variance (ANOVA) on the mean response after drug and capsaicin injection; genotype (control vs. Agrp^Trpv1) and drug (vehicle vs. CL 316,243) were used as factors for the ANOVA. Holm–Sidak's multiple comparisons test (MCT) was used to find post hoc differences among groups. MCTs are indicated as ***$P < 0.001$, and ****$P < 0.0001$ in figure panels. Control mice + vehicle ($n = 9$); control mice + CL 316,243 ($n = 9$); Agrp^Trpv1 mice + vehicle ($n = 5$); Agrp^Trpv1 mice + CL 316,243 ($n = 6$). Dashed line indicates time of injections. Colored symbols indicate mean ± SEM. Gray symbols indicate individual values. **k** Dose response of CL 316,243 injected immediately before capsaicin in Agrp^Trpv1 mice (dashed line denotes injection time). Number of animals used per experimental group shown in the panel. **l** Food intake response to activation of Agrp neurons when injected with different doses of CL 316,243 ($n = 7$ for all groups); interaction ($F_{2, 36} = 0.95$, $P = 0.39$); drug ($F_{2, 36} = 4.16$, $P = 0.02$); genotype ($F_{1, 36} = 64.96$, $P < 0.0001$). Statistics: two-way ANOVA with genotype and drug as factors. Holm–Sidak's multiple comparisons test (MCT) was used and indicated as ***$P < 0.001$ and ****$P < 0.0001$. **m** Related to **l**, but fold change in food intake in Agrp^Trpv1 related to control mice. Bars and symbols indicate mean ± SEM

biochemical analysis and the brain to confirm ablation of Agrp neurons. After diphtheria toxin treatment, only minimal residual Agrp-positive neuronal fibers were visualized in the arcuate nucleus (Fig. 7d), confirming the ablation of Agrp neurons is not affected by enteral feeding. In the WAT, we measure genes involved in lipid metabolism and found decreased expression of several genes involved in lipogenesis (Fig. 7e). Altogether, these results support the idea that Agrp neuron activity mediates weight gain and adiposity in positive energy balance conditions.

**Discussion**
Here, we reported that Agrp neurons shift metabolism towards lipid storage, a mechanism we suggest being important for fat

accumulation during positive energy balance (i.e., a metabolic state coupled to weight gain).

During diet-induced obesity, the activity of Agrp neurons is elevated as recorded using slice electrophysiology[32–35]. This elevated activity of Agrp neurons could be involved in the metabolic shifts towards fat deposition (lipogenesis) reported here. In fact, a recent report showed that activation of Agrp neurons in Agrp^hM3Dq mice during 2 weeks by providing clozapine-N-oxide (CNO)—the pharmacological agonist of the excitatory receptor hM3Dq—in the drinking water increased food intake only during the first 2 days[36]. However, mice continued to increase body weight during the entire period of study, suggesting a role for Agrp neurons in body weight gain that was not due to increased feeding[36]. Additionally, mice with impaired activity of Agrp

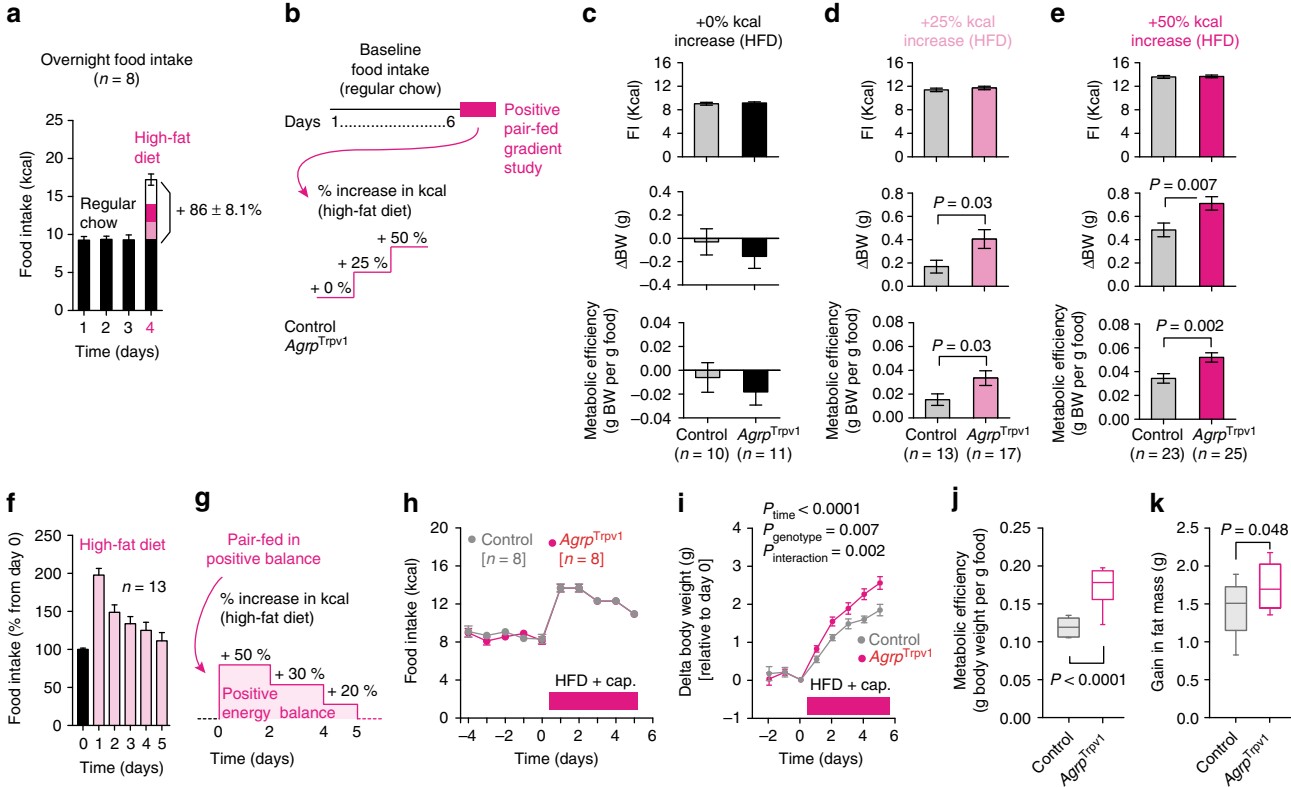

**Fig. 6** Agrp neuron activity boosts fat gain in obesogenic conditions. **a** Physiological hyperphagia after acute switch to high-fat diet (45% kcal from fat) in mice. **b** Pair feeding studies: high-fat diet (0, 25 or 50% more calories compared to ad libitum normal chow food intake) was provided after injection of control and Agrp$^{Trpv1}$ mice with capsaicin. **c** Top, Mice were fed the same number of calories as they ate during baseline food intake; Middle, delta body weight; Bottom, metabolic efficiency. **d** Similar to **c** but in mice fed 25% more calories when switched to a high-fat diet. **e** Similar to **c** and **d**, but in mice fed 50% more calories. **f** Food intake measured during 5 days after switching mice to a high-fat diet (45% kcal from fat); in pink, period of dietary switch. **g** Control and Agrp$^{Trpv1}$ mice were switched to a high-fat diet (shaded pink) after baseline food intake measurements; capsaicin was injected every day before dark cycle. **h** Food intake. **i** Delta changes in body weight relative to the day of diet switch. **j** Metabolic efficiency and **k** gain in fat mass during the period in which mice were fed high-fat diet. Unpaired t-test was used in **c**, **d**, **e**, **j** and **k**. Two-way analysis of variance (ANOVA with time as a repeated measure was used in **i**. Number of mice is displayed in the figures. Bars and symbols indicate mean ± SEM. Boxes indicate median ± 25/75 quartiles ± min/max values. Statistically significant P values are provided in the figures

neurons during high-fat feeding were resistant to diet-induced obesity with only minor or no effects on food intake[32]. Thus, previous reports and our current findings strongly suggest that persistent elevation of Agrp neuron activity during positive energy balance propagate adiposity.

In the WAT, activation of the sympathetic nervous system leads to lipolysis[37–39], whereas its inhibition favors lipogenesis[40]. Our results suggest a model in which Agrp neuron activity during obesogenic conditions suppresses sympathetic tone in the WAT to promote adiposity. In support of this model, activation of Agrp neurons decreased the sympathetic nervous system activity in peripheral tissues[19,20,41]. Direct recordings of sympathetic nerve electrical activity showed that activation of Agrp neurons decreased in approximately 30% the firing rate of sympathetic nerves[20,41]. Conversely, inhibition of Agrp neurons increased sympathetic nerve activity in up to 60%[41]. These effects were mediated by projections of Agrp neurons to paraventricular nucleus (PVH) and dorsomedial hypothalamic nucleus[41]. Together with our studies using a pharmacological agonist of β3-adrenergic receptors (Fig. 4), these findings support that Agrp neurons control sympathetic nerve activity in the WAT to control lipogenesis and adiposity. As we will discuss below, we speculate that in addition to diet-induced obesity this function of Agrp neurons is important in the daily anticipatory responses to food ingestion[42].

Agrp neurons release gamma-aminobutyric acid (GABA)[43,44] and neuropeptide-Y (NPY)[2,43], in addition to AGRP[2,4]. In our experiments, we did not identify which of these transmitters is involved in the effects of Agrp neurons on peripheral substrate utilization. However, previous reports showed that infusion of NPY in the rat brain increases RER[45–47]. The effect of NPY on substrate utilization was prominent in the PVH[47], an area in which NPY treatment also induces hyperphagia[48,49]. Because Agrp neurons project to the PVH to promote feeding[50,51], it is possible that NPY released from Agrp neurons in the PVH also promotes the observed changes in substrate utilization. In addition to NPY, Agrp neurons also likely regulate lipogenesis via release of the neuropeptide AGRP, which is an endogenous antagonist of melanocortin receptors[3,4,52–54]. Pharmacological inhibition of melanocortin receptors in the brain—by delivery of SHU9119—increased adiposity and RER[55], similar to our results. The pharmacological agonist of melanocortin receptors, MTII, when infused in the rat brain increased the sympathetic nervous system activity in the WAT, an effect blocked by SHU9119[55]. Thus, it is plausible that Agrp neurons regulate substrate utilization and lipogenesis via release of both NPY and AGRP.

Contrary to adult ablation of Agrp neurons which leads to aphagia[17,18,56], neonatal ablation of these neurons is compatible with life[17]. In a previous study[57], Agrp neurons were ablated during the first postnatal week. The authors found that adult mice

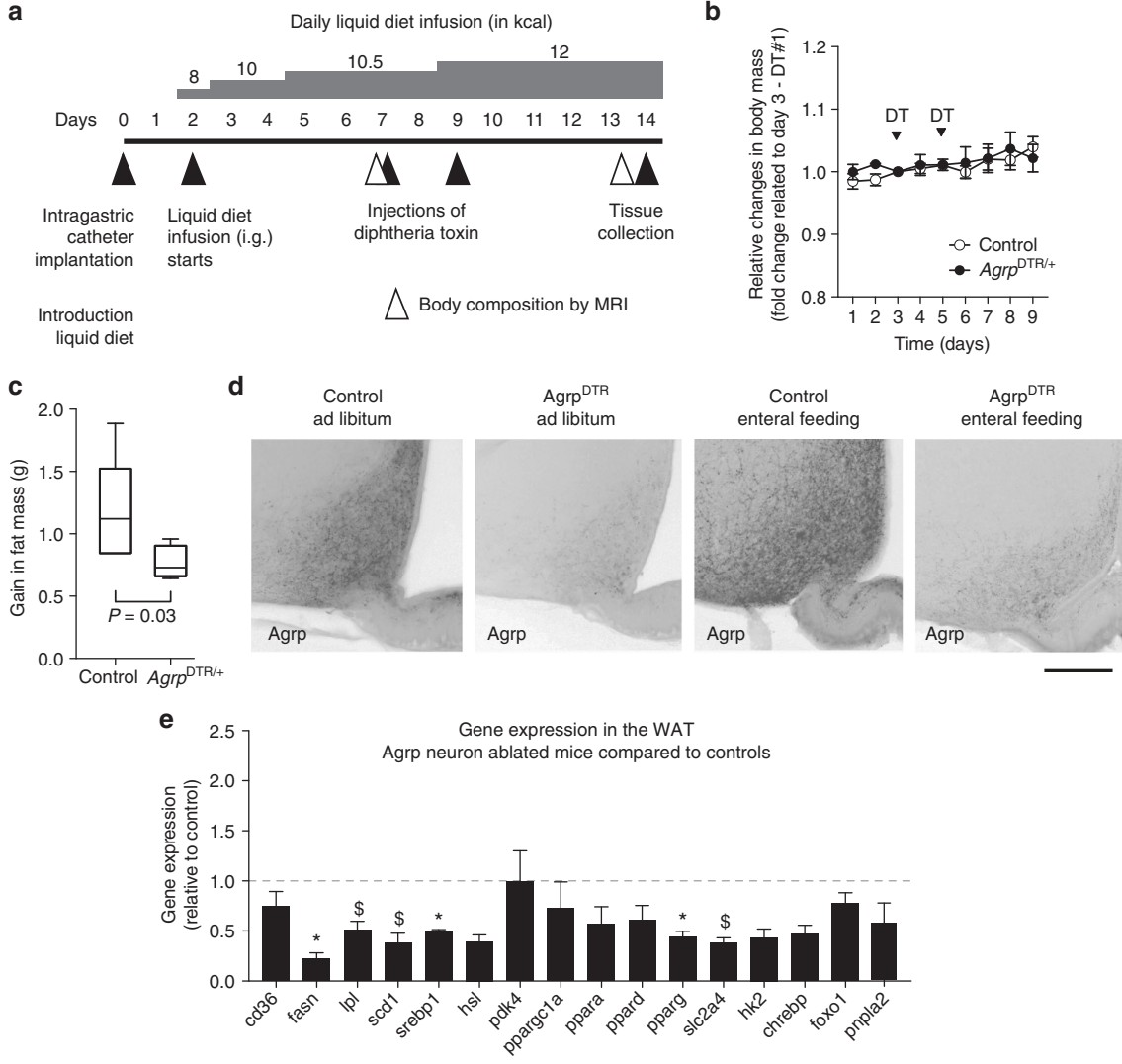

**Fig. 7** Agrp neuron ablation impairs fat gain during positive energy balance. **a** Model of positive energy balance by intragastric delivery of liquid diet in mice concomitant with ablation of Agrp neurons. **b** Relative changes in body weight represented by fold-changes related to day 3 of the protocol—when animals received the first injection of diphtheria toxin. **c** Gain in fat mass. **d** Immunohistochemistry for AGRP. **e** Gene expression in the WAT. Data are normalized to control levels (dashed line). Statistical analysis was performed using Student's $t$-test. *$P < 0.05$, $^{\$}P < 0.10$ (non-significant). Bars and symbols indicate mean ± SEM. $N = 4$ animals per group. Boxes indicate median ± 25/75 quartiles ± min/max values. $P$ values are provided in the figures. Scale bar = 50 μm

without Agrp neurons showed decreased RER at the beginning of the dark cycle[57], a time when Agrp neuron activity is high[11]. These results suggested that Agrp neurons are important for the increase in RER that occurs at the onset of the dark phase when lipogenesis is turned on and animals enter an anabolic state, in anticipation to food ingestion. In support of this argument, animals trained in a scheduled feeding regimen showed an elevation of NPY levels in anticipation to a meal[58,59] and NPY signaling promotes lipogenesis[60]. Additionally, when fed a scheduled feeding regimen, mice with partial ablation of Agrp neurons showed impaired anticipatory activity before food presentation[61]. Thus, these findings support a function for Agrp neurons in coordinating the daily anticipatory responses to food intake (see also ref. [42]), including anticipatory substrate utilization switches.

More recently, two additional reports highlighted the involvement of Agrp neurons in peripheral substrate metabolism[62,63]. These studies reported on the role of carnitine acetyltransferase (Crat) as a critical enzyme in Agrp neurons to control the switch in peripheral energy metabolism during different metabolic states. Mice knockout for Crat selectively in Agrp neurons had impaired

metabolic flexibility—the capacity to shift between energy substrates—and were more predisposed to fat accumulation during positive energy balance[62,63]. However, the authors did not directly assess the activity of Agrp neurons. Based on the experiments reported here, an increase in Agrp neuron activity in Cart knockout Agrp neurons is expected during positive energy balance. Collectively, these recent reports support the importance of Agrp neurons in regulating peripheral substrate utilization and fat accumulation.

In summary, we showed that Agrp neurons in the hypothalamus rapidly control whole-body nutrient utilization by shifting metabolism towards lipogenesis. These findings have implications for our understanding of how neuronal circuits control energy metabolism and, consequently, to our understanding of severe disordered conditions such as obesity.

## Methods
**Animals**. Mice used in the experiments were 3 to 8 months old from both genders. $Agrp^{Trpv1}$ mice were generated by crossing $Agrp^{Cre}$ to $Rosa26^{LSL-Trpv1}$ mice backcrossed to a $Trpv1^{KO}$ background to prevent capsaicin action on other cell

types[64,65]. As a result, $Agrp^{Trpv1}$ mice selectively expressed the capsaicin-sensitive channel, Trpv1, in Agrp neurons. Because capsaicin is a highly specific ligand of Trpv1[66], peripheral injection of capsaicin allowed for a rapid, reliable and transient chemogenetic activation of Agrp neurons in $Agrp^{Trpv1}$ mice[14,19]. Using $Agrp^{Trpv1}$ mice allowed us to rapidly activate Agrp neurons without the necessity of tethers (as for example, using optogenetics), facilitating the study of animals in indirect calorimetry chambers to measure gas ($O_2$ and $CO_2$) exchange. Importantly, Trpv1-mediated activation of Agrp neurons was both rapid (latency to start is ~2 min) and short lived (last <1 h)[65], unlike the effects of activating Agrp neurons via the designer receptor hM3Dq[16].

More specifically, $Agrp^{Trpv1}$ mice were: $Agrp^{Cre/+}::Trpv1^{KO/KO}::R26\text{-}LSL\text{-}Trpv1^{Gt/+}$; control animals were either $Agrp^{Trpv1}$ mice injected with vehicle (3.3% Tween-80 in saline) or $Trpv1^{KO/KO}:R26\text{-}LSL\text{-}Trpv1^{Gt/+}$ mice injected with capsaicin. In the experiments reported, all mice were littermates (Agrp neuron activated and controls). We did not observe any differences between the two control groups (in feeding behavior and calorimetry measurements[14,19]). Therefore, throughout the study we referred to both groups as "controls". We have carefully characterized this animal model elsewhere[14,19]. Briefly, to control for ectopic expression of Cre outside Agrp neurons, we have genotyped all our animals to the excised conditional allele[14] and found only rare (<1%) occurrence of ectopic excision of $Rosa26^{LSL\text{-}Trpv1}$. Mice with ectopic excision of the conditional allele were excluded from our studies.

We have performed dose–response curves for capsaicin and identified the dose of 10 mg kg$^{-1}$ (intraperitoneal (i.p.)) as optimal to induce feeding behavior in $Agrp^{Trpv1}$ mice. We have also performed a dose response of capsaicin and measured changes in RER (1, 3, 10 and 30 mg kg$^{-1}$, i.p.; experiments not reported). We found that 10 mg kg$^{-1}$ was also the optimal dose to promote changes in RER. Thus, we selected this dose of capsaicin for our studies. The following mouse lines were used in this study: $Agrp^{tm1(cre)Lowl/J}$ (stock number 012899, Jax); $Gt(ROSA)26Sortm1(Trpv1,ECFP)Mde/J$ (stock number 008513, Jax); $Trpv1^{tm1Jul/J}$ (stock number 003770, Jax); and $Agrp^{DTR}$ (donated by Richard Palmiter). All animals were kept in temperature- and humidity-controlled rooms, in a 12/12 h light/dark cycle, with lights on from 7:00 AM to 7:00 PM. Food and water were provided ad libitum, unless otherwise stated. All procedures were approved by Institutional Animal Care and Use Committee (Yale University).

**Drugs**. The following compounds were used in the reported studies: CL 316,241 (in saline; from Tocris); C75 (RPMI medium 1640; from Tocris);[67,68] capsaicin (3.3% Tween-80 in saline; from Sigma)[14], CNO (in saline; from Enzo Life Science)[14], and diphtheria toxin (in saline; List Biological, cat. no. 150). All drugs were injected in a volume of 10 ml kg$^{-1}$ of body weight i.p. except for diphtheria toxin, which was injected at the dose of 50 μg kg$^{-1}$ and the volume was 2.66 μl g$^{-1}$ of body weight. When multiple injections were performed in the same experiment, the volume of each injection was adjusted to a total volume of 10 ml kg$^{-1}$ per animal.

**Viral vector and stereotaxic surgery**. The 3–5-month-old homozygous Agrp-IRES-Cre mice ($Agrp^{tm1(cre)Lowl/J}$) were used for stereotaxic viral injections. At 30 min prior to the first incision, animals were administered a subcutaneous injection of the analgesic buprenorphine (0.1 mg kg$^{-1}$). Animals were anesthetized with a ketamine/xylazine cocktail (100 and 10 mg kg$^{-1}$, respectively) and placed upon a heated stage to await a sufficient plane of anesthesia. Next, a small amount of ophthalmic ointment was applied to the eyes to protect from drying. The stage was then situated within a stereotaxic instrument with digital display (Kopf model #942) and the animal was head-fixed with ear bars. After exposure of the skull via a small incision, the bregma and lambda points were identified and the head was leveled in an anterior–posterior fashion by adjusting these two points to the same z-depth. Lateral leveling was performed by choosing a point at a fixed lateral distance from bregma and the head was adjusted to match the z-depth on each side. Two small holes were drilled in the skull above the points of injection using a 0.9 mm diameter carbon steel burr attached to a high speed stereotaxic drill (Kopf Model #1474). The arcuate nucleus of the hypothalamus was targeted utilizing the following coordinates with bregma as the reference point: anterior/posterior: –1.4 mm, lateral: ± 0.3 mm, dorsal/ventral: −5.9 mm. The virus rAAV5-hSyn-DIO-hM3D(Gq)-mCherry (University of North Carolina Vector Core, titer 6.0 × 10$^{12}$ viral genomes per ml) was then loaded into a blunt tip Neuros syringe (1 μl, Hamilton, cat. no. 7001KH) and the needle was slowly lowered into place. Then, 500 nL bilateral injections of the virus were administered at a rate of 100 nL min$^{-1}$. The needle was left in place for 6 min post injection to allow for spread of the virus, then was slowly withdrawn to prevent backflow of the virus up the needle track. Skin above the skull was closed with two Michel suture clips and the animal was administered a subcutaneous injection of sterile saline and allowed to recover on a heated pad. As a post-operative analgesic, buprenorphine (0.1 mg kg$^{-1}$) was administered every 6–12 h after surgery for a total of 48 h. Mice were allowed to recover within their homecage and testing commenced at a minimum of 3 weeks after injection to allow for sufficient infection and viral gene expression.

**Food intake assay**. Mice injected with rAAV5-Ef1a-DIO-hM3D(Gq)-mCherry were singly housed 1 week prior to food intake studies. Animals were administered saline (i.p.) every other day for a total of 3 injections to acclimate to handling and injection. At 48 h prior to the test, mice were placed in a fresh cage to minimize any food that might be in the bedding. On the day of the test, mice were injected with either saline or CNO and food intake was manually assessed after injection.

**Intragastric surgery**. The 8–10-week-old singly housed $Agrp^{DTR/+}$ and $Agrp^{+/+}$ mice were used. At 30 min prior to the first incision, animals were administered a subcutaneous injection of the analgesic buprenorphine (0.1 mg kg$^{-1}$). Inhalation anesthesia was initiated in an induction chamber via an isofluorane vaporizer. After being deeply anesthetized, animals were immediately laid on a heated mat and a nosecone delivering 2 % isofluorane was placed over the nose of the animal to provide continuous plane of anesthesia throughout the procedure. The dorsal neck and abdomen were shaved and then cleaned with alternating scrubs of 70% ethanol and betadine. A dorsal midline skin incision was made on the neck of the animal and the skin immediately surrounding the incision was separated from the muscle with scissors. Also, a subcutaneous tunnel was opened by extending this separation toward the right flank. Following this, an approximately 2 cm vertical midline abdominal incision was made in the skin and scissors were used to separate the skin from the muscle surrounding the incision and to the right side of the animal. A 1.5 cm incision was then made in the linea alba to allow access to the abdominal cavity. The stomach was gently maneuvered, with sterile cotton swabs saturated with saline, to give specific access to the forestomach. A small triangular pattern of 3 stitches was made with 4-0 silk sutures in the forestomach leaving loose ends of the suture to allow for cinching after insertion of the infusion tube. A 13 cm piece of Micro-Renethane tubing (Braintree Scientific, MRE065, 0.065" OD × 0.030" ID) was previously prepared by quickly exposing the tip to a flame to round the edge and provide a subtle flange. The non-flanged end was connected to a 1 mL syringe containing sterile saline. Using a 20 G sterile needle, the stomach was carefully punctured in the center of the triangular stitch and the flanged end of the tube was inserted. The silk suture was then cinched closed and tied. The tubing was gently pulled to verify secure attachment to the stomach and 0.1 mL of saline was infused to confirm flow and a leak free connection. Prior to any subsequent removal of the syringe attached side of the tubing, the tubing was clamped with a soft hemostat to prevent any leakage from the stomach. To route the tube through the peritoneum, a sharp forceps was used to puncture the right exterior side of the peritoneum and the tube was inserted over the tip and pulled through the hole. Then, allowing ample slack between the peritoneum and the stomach, the tube was sutured to the abdominal wall using 4-0 silk. This prevents slippage of the tubing during normal movement. The abdominal wall was then closed with 4-0 absorbable Vicryl sutures in a continuous pattern. The animal was laid on its side and a trochar was inserted through the skin on the neck and routed dorsally and then laterally until it could be visualized close to the site of the tube. The tube was then inserted into the trochar and pulled through the incision on the back of the neck. The abdominal skin was closed with 5-0 Monofilament sutures. A sterile polyester felt button, 0.563 in. (14 mm) diameter (Instech, cat. no. DF65BS) was moistened with saline and inserted over the tubing and placed under the skin on the interscapular region of the animal. This procedure allows for secure externalization of the tubing while preventing stress to the skin. The skin was then closed over the button with 5-0 monofilament sutures and a small droplet of tissue adhesive was used to secure the tube to the plastic exit hole on the felt button. The tubing was cut 12 mm above the button and a 5 mm blunted and crimped section of an 18 G needle was inserted into the tube to keep the line clean and to prevent any leakage of gastric liquid up the tube. Animals were then given a subcutaneous injection of sterile saline and allowed to recover from anesthesia on a heated pad, then returned to their homecage. As a post-operative analgesic, buprenorphine (0.1 mg kg$^{-1}$) was administered every 6–12 h after surgery for a total of 48 h.

**Intragastric diet infusion**. Immediately after returning to the homecage, mice were provided with ad libitum access to water, normal chow (Envigo, Teklad 2018S), and a nutritionally assayed AIN-76 liquid diet (Bio-Serv, cat. no. F1268; composition: protein, 180 kcal L$^{-1}$; fat, 125 kcal L$^{-1}$; carbohydrate, 695 kcal L$^{-1}$—total of 1 kcal mL$^{-1}$). Liquid diet, supplied as a powder, was mixed fresh daily according to the manufacturer's specifications. After passing through a mesh filter, liquid diet was provided to mice in 50 mL glass feeding tubes (Bio-Serv cat. no. 9019) mounted to the side of the homecage. At 2 days post surgery, normal chow was removed, and mice were connected to a 1.5 m length of Micro-Renethane tubing attached to syringes containing liquid diet. Syringes were mounted in a syringe pump (Harvard Apparatus, model PHD2000) and diet was infused at the following volumes: day 2: 8 mL at 0.4 mL h$^{-1}$; days 3–4: 10 mL at 0.46 mL h$^{-1}$; days 5–8: 10.5 mL at 0.48 mL h$^{-1}$; days 9–14: 12 mL at 1.84 mL h$^{-1}$. Every 24 h, the 1.5 m tube was detached, and animals were administered 0.1 mL of saline through the catheter in order to flush the line and keep it clean. On post-operative days 7 and 9, mice were administered a 50 μg kg$^{-1}$ intramuscular injection of diphtheria toxin. On days 7 and 13 postoperatively, mice were assessed for body composition in an Echo-MRI 100H analyzer. Animals were killed on day 14 for tissue collection.

**Metabolic assays and biochemical analysis**. Blood samples were collected from the tail in order to measure glucose and free fatty acids (NEFA) levels. Glucose was measured using a One Touch Ultra 2 glucometer. After blood centrifugation,

serum was collected and used to measure NEFA as indicated by the manufacturer (WAKO, Japan). Body composition was assessed using an Echo-MRI system. Fat gain was measured by the delta between the last and the first MRI scan.

**Gene expression and western blotting**. Animals were deeply anesthetized with ketamine and xylazine and killed by decapitation. Tissues were collected and frozen in liquid nitrogen. Tissues were lysed in buffer containing 1% Nonidet P-40, 50 mM Tris 3 HCl, 0.1 mM EDTA, 150 mM NaCl, proteinase inhibitors and protein phosphatase inhibitors. Equal amounts of protein lysate were electrophoresed on sodium dodecyl sulfate–polyacrylamide gel electrophoresis and transferred to polyvinylidene difluoride membranes. Primary antibodies (Lipolysis Activation Antibody Sampler Kit #8334, Cell Signaling) were incubated at 4 °C overnight at a 1:1000 dilution. Membranes were washed and incubated with secondary antibodies conjugated to horseradish peroxidase. Protein levels were visualized using ECL chemiluminescent substrate and quantified using ImageJ.

Total RNA was extracted from mouse tissues using RNeasy® lipid mini kit (Qiagen). Complementary DNA was reverse transcribed (Bio-Rad) and amplified with SYBR Green Supermix (Bio-Rad) using a Light Cycler 480 real-time PCR system (Roche). Data were normalized to the expression of *Actin*, *Gusb* and *Arbp*. The following primers were used: *cd36* (F-GCGACATGATTAATGGCACA; R-CCTGCAAATGTCAGAGGAAA); *fasn* (F-GTCGTCTGCCTCCAGAGC; R-GTTGGCCCAGAACTCCTGTA); *lpl* (F-TTCACTTTTCTGGGACTGAGAATG; R-GCCACTGTGCCGTACAGAGA); *scd1* (F-CCGGAGACCCCTTAGATCGA; R-TAGCCTGTAAAAGATTTCTGCAAACC); *srebp1* (F-GGCTCTGGAACAGACACTGG; R-TGGTTGTTGATGAGCTGGAG); *hsl* (F-CCTGCAAGAGTATGTCACGC; R-GGAGAGAGTCTGCAGGAACG); *cpt1b* (F-TGTCTACCTCCGAAGCAGGA; R-GCTGCTTGCACATTTGTGTT); *pdk4* (F-TGACAGGGCTTTCTGGTCTT; R-AGTGAACACTCCTTCGGTGC); *ppargc1a* (F-TGTAGCGACCAATCGGAAAT; R-TGAGGACCGCTAGCAAGTTT); *ppara* (F-AGTTCGGGAACAAGACGTTG; R-CAGTGGGGAGAGGAGGAACA); *ppard* (F- ACTCAGAGGCTCCTGCTCAC; R-GGTCATAGCTCTGCCACCAT); *pparg* (F-GATGCACTGCCTATGAGCAC; R-TCTTCCATCACGGAGAGGTC); *slc2a4* (F- CAGTGTTCCAGTCACTCGCT; R-TTTTAAAACAAGATGCCGTCG); *hk1* (F-GCGAGGACAGGCTGTAGATG; R-GAATTTCATCAGAGAGCCGC); *hk2* (F- GGAGCTCAACCAAAACCAAG; R-GGAACCGCCTAGAAATCTCC); *chrebp* (F-CTGGGGACCTAAACAGGAGC; R-GAAGCCACCCTATAGCTCCC); *foxo1* (F- GCGCATAGCACCAAGTCTTCA; R-AGCGTGACACAGGGCATCA); *actb* (F-GGCTGTATTCCCCTCCATCG; R-CCAGTTGGTAACAATGCCATGT); *gusb* (F- CCGACCTCTCGAACAACCG; R- GCTTCCCGTTCATACCACACC); *arbp* (F-CGACCTGGAAGTCCAACTAC; R-ATCTGCTGCATCTGCTTG). Additionally, the TaqMan probe for *pnpla2* was used (Mm00503040_m1).

**Immunohistochemistry**. Mice were deeply anesthetized by isoflurane inhalation and killed by decapitation. Brains were immersion fixed in 4% paraformaldehyde for 48 h at 4 °C. Next, coronal brain sections of the arcuate nucleus were cut at thickness of 50 μm on a vibrating blade microtome (Leica VT1000S) and free-floating tissue sections were washed in phosphate-buffer saline (PBS, pH 7.4). Sections were incubated in 0.3% Triton X-100 in PBS for 30 min at room temperature for permeabilization. For blocking, sections were incubated in PBS containing 5% normal donkey serum and 0.3% Triton X-100 for 60 min at room temperature. Tissue was then treated with the primary antibody, Rabbit-anti-AgRP (Pheonix Pharmaceuticals, cat. H-003-57, 1:1000), in blocking solution overnight at room temperature. Tissue sections were next washed 5 times for 5 min in PBS and subsequently incubated in PBS containing the secondary antibody, Donkey-anti-Rabbit Alexa Fluor 594 (Invitrogen, A21207), for 2 h at room temperature. Finally, sections of the arcuate nucleus were washed in PBS, mounted and coverslipped with ProLong Gold Antifade Reagent (Invitrogen) for fluorescent microscopic analysis.

**Indirect calorimetry and nutrient utilization**. Nutrient utilization can be measured by indirect calorimetry, where the measurements of $VCO_2$ production and $VO_2$ consumption are used to calculate the RER[23]. Under normal conditions, a RER approaching 0.7 indicates predominant fat oxidation, while a RER approaching 1.0 indicates predominant carbohydrate oxidation[23]. Oxygen consumption ($VO_2$) and $CO_2$ production ($VCO_2$) were measured in four to eight mice simultaneously in indirect calorimetry chambers (TSE Systems, Germany). Measurements were recorded every 8–12 min over the entire course of the experiment (except for the experiment reported in Fig. 1 and Supplementary Figure 1, in which measurements were recorded every 30 min). RER was calculated as the ratio between $VCO_2$ and $VO_2$. Whole-body fat utilization was calculated using the follow equation: $1.67 \times (VO_2 - VCO_2)$. Whole-body carbohydrate utilization was calculate using the follow equation: $4.55 \times VCO_2 - 3.21 \times VO_2$[23]. In these equations, $VO_2$ and $VCO_2$ were in $L\,min^{-1}$ and constants are in $L\,g^{-1}$ (representing the amount of gas production or consumption in liters per gram of substrate oxidized. The calculate substrate utilization rate (in $g\,min^{-1}$) was converted to $mg\,min^{-1}$ for better visualization. All animals were single housed during the experiments in calorimetry chambers. Mice were acclimated to the chambers for several days until they displayed normal circadian variance in metabolic measurements (as reported

in Supplementary Figure 1). Typically, this acclimation period lasts ~5–7 days in our laboratory. We also acclimated animals to mock injections to minimize stress response. In all experiments, animals were injected 3–4 times with vehicle in consecutive days (Supplementary Figure 1).

For the glucose response study, mice were provided with glucose (D-Glucose, G8270, Sigma, USA) via gavage. We used saline as vehicle and three doses of glucose (1, 2 and 3 g kg⁻¹ body weight, via gavage feeding). Food was removed 2 h before the experiment during the light cycle of the animals. Indirect calorimetry was recorded for 60 min prior injection of capsaicin and diet switch. For the experiments where no food was provided, food was removed from the cages 2 h before injecting mice with capsaicin/vehicle. Baseline indirect calorimetry was recorded for 60 min, and then the effects of Agrp neuron activation were recorded for additional 60–120 min. Similar procedures were used in the experiments using hM3Dq to activate Agrp neurons and in the experiments in which compounds were given to mice prior the experiment. Glucose (2 g kg⁻¹, via gavage), C75 (10 mg kg⁻¹, i.p.) or CL 316,243 (0.01-1.00 mg kg⁻¹, i.p.) were given together with capsaicin. In all cases, the drugs were injected immediately before capsaicin using two different syringes.

**Statistical considerations**. Matlab R2016a, PASW Statistics 18.0, Prism 7.0 and Adobe Illustrator CS6/CC were used to analyze data and plot figures. Student's *t*-test was used to compare two groups. Analysis of variance (ANOVA) was used to compare multiple groups. When necessary, multiple comparisons post hoc test (MCT) was used (Holm–Sidak's test). When homogeneity was not assumed, the Kruskal–Wallis non-parametric ANOVA was selected for multiple statistical comparisons. The Mann–Whitney *U* test was used to determine significance between groups. Statistical data are provided in the figures. $P < 0.05$ was considered statistically significant.

**Reporting summary**. Further information on experimental design is available in the Nature Research Reporting Summary linked to this article.

## Data availability

The data that support the findings of this study are available from the corresponding author upon reasonable request.

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

## Acknowledgements

We thank Serge Luquet, Matthew Rodeheffer, Ivan de Araujo, Hai-bin Ruan and Xiaoyong Yang for comments on this project and manuscript. Serge Luquet raised the

important point that Agrp neurons mediate the physiological increase in RER in anticipation to ingestion. M.O.D. received support from Brain and Behavior Research Foundation (NARSAD Young Investigator Award), Yale Center for Clinical Investigation Scholars Award (NCATS, UL1 TR000142), National Institute of Diabetes and Digestive and Kidney Diseases (1R01DK10791601), DRC (P30 DK045735), Whitehall Foundation, Charles H. Hood Foundation, CNPq (487096/2013-4 and 401476/2012-0, Brazil) and CAPES (88881.068059/2014-01, Brazil). J.P.C.-d.-A. and M.R.Z. were partially supported by a fellowship from the Science Without Borders program (Brazil).

## Author contributions

J.B. performed experiments and analyzed data. M.R.Z. generated critical animal models and performed initial experiments. J.P.C.-d.-A. designed and performed experiments and analyzed data. M.O.D. designed and performed the experiments, analyzed the data, and wrote the manuscript.

## Additional information

**Competing interests:** The authors declare no competing interests.

