## [Peer Review File · Nature Communications]

Reviewers' comments:

Reviewer #1 (Remarks to the Author):

This manuscript by Dr. Dietrich and colleagues reports novel observations that transient, chemogenetic activation of AgRP neurons rapidly and acutely alters whole body nutrient utilization measured using indirect calorimetry, independent of changes in energy expenditure. In fed mice with food withheld during the experiment, some changes in lipogenic/lipolytic genes/protein were observed in adipose tissue suggesting increased lipogenesis with acute AgRP neuron activation. AgRP neuron activation also increased body weight gain and modestly increased fat mass gain in mice given a controlled high fat feeding regime for 5 days. AgRP neuron activation was induced via ip capsaicin injection in mice that express capsaicin-sensitive channel Trpv1 in AgRP neurons, a mouse model the authors previously characterized (Dietrich et al, Cell 2015; Ruan et al, Cell 2014). This work extends knowledge from other studies that demonstrate that AgRP neuron activation can affect metabolism beyond its classical role on feeding behaviour.

The manuscript is well written with a coherent progression of ideas. The authors are commended for the work performed. The impact of this manuscript is limited by the following concerns:

1. A widely known function of activated hypothalamic AgRP neurons to stimulate feeding behaviour in both fasted and sated states. Interestingly, however, one of the main claims is that AgRP neuron activation alters nutrient utilization "independent of food ingestion" (Introduction, p3, last line), or variations of the same [e.g. "absence of ingestive behaviours (Abstract), etc.]. The authors have provided references to other studies in which AgRP neuron activation has metabolic effects beyond feeding, but I am not convinced that the studies herein fully exclude alterations in feeding behaviour induced by AgRP neuron activation. Fig 2 shows an effect of AgRP neuron activation on changing substrate utilization with and without glucose gavage in fed mice (presumably with no access to food; not clear in Methods), and the mean effect being greater in AgRP-Trpv1 mice with glucose gavage than control counterparts. However, could that moderate increase be due in part to increased food seeking activity of the mice? In the absence of food/glucose, fed mice with AgRP neuron activation show increased activity (Suppl Fig 3f), possibly Agrp-stimulated food seeking behaviour, and as the authors also state, physical activity can increase RER. Interestingly, this increase in activity coincides at ~30 min (Fig 3f), the time in which the added effect of AgRP activation is seen in Fig 2. In Figure 1 in the presence of food, how much are the mice eating during the time course of the experiment? The increase in carbohydrate utilization in the presence of food appears much more prolonged (sustained for few hours) compared experiments in the absence of food, in which carbohydrate utilization peaks at t=30 mins. Food intake data for Fig 1 experiments would also be useful.

2. Results – Figure 1 – Authors state that RER is calculated from dividing VCO₂ by VO₂ (Methods, p16). If there is no differences between AgRP-TRPV1 and control mice in VCO₂ (Fig 1c) or VO₂ (Fig 1d)s why is RER so drastically different between the 2 groups?

3. Blood glucose levels are shown in Fig 3C – could blood glucose data also be provided in glucose gavage experiments (Fig 2 and Suppl Fig 2)? Blood glucose profiles could be useful in discussing nutrient utilization results.

4. In Ref 24, one of the metabolic effects of activation of AgRP neurons was inhibition sympathetic nerve activity (SNA), at least to brown adipose. Inhibition of SNA would also decrease lipolysis, which would also lead to decreased blood NEFA levels. Inhibition of SNA could also decrease HSL activity, which agrees with HSL western data (Fig 3e). The involvement of AgRP neuron activation and the sympathetic nervous system could be touched on in Discussion.

5. The changes induced by AgRP neuron activation on pHS� protein levels and gene expression of hk2 and ppar-alpha in WAT as indicators of promoting lipogenesis is strengthened by experiments

using FAS inhibitor C75. However, a stronger piece of evidence is simply fat mass. In normal chow fed mice (e.g. not HFD data in Fig 4), is there an increase in adiposity, particularly given the lasting effects of AgRP stimulation as shown in Fig 1? Lipogenesis would be expected to be lower in mice during the light cycle, during which AgRP neurons were activated, thus may help to amplify changes in lipogenesis.

6. Experiments in Fig 4 are nicely designed, but a few concerns arise: (a) HFD was given for 5 days, but body weight data for the 5th day (Fig 4i) is missing. Please include day 5 data. (b) What is the axis showing in Fig 4i – is it BW fold increase? Please clarify. (c) How many days following HFD was data for Fig 4j and 4k obtained? (d) In this step-wise increase of HFD feeding for 5 days, do mice become more “obese”, compared to regular chow fed mice, even without chemogenetic activation of AgRP neurons? If yes, then would AgRP neurons be intrinsically activated (by DIO, as previously shown), and what then is the added stimulation of AgRP neurons? For example, in this DIO model, capsaicin is still injected daily to activate AgRP. (e) Fig 4k represents very modest changes differences in fat mass gain between the 2 groups. Is this difference in fat mass gain sufficient to explain the difference in body weight changes?

7. The title of the manuscript and elsewhere in the text suggests that the hypothalamus is the specific brain region involved in the metabolic effects demonstrated. Whereas AgRP neurons are well known to be found in specific brain regions to control energy metabolism (e.g. ARC), do the authors have evidence in their model that only the hypothalamus is affected?

8. Suppl Fig 4 – AAV-DIO-hM3D(Gq)mCherry mice are not found in the Methods. Please include more details. Additionally, have these mice been validated elsewhere? This figure is missing control mice – what is the effect of CNO alone, and why is CNO action on its receptor hM3D(q) in AgRP neurons indicative of neuronal activation? Are HM3D ubiquitously expressed? Readers would benefit from additional information about this model if used in the present study.

Minor Considerations

9. Methods – p16 – Formulas for fat utilization and carbohydrate utilization are missing units. Please provide units for the constants used in the formula.

10. Methods – p15 – In the subsection, “Drugs”, please provide references for the reported studies indicated in the text.

11. Discussion – p11 – “elevated activity of AgRP neurons” should be changed to “activation of AgRP neurons”.

12. Discussion – 11 – ref 43 is a submitted work and not published and should perhaps then be noted as “unpublished data” unless status of submission is changed.

13. Discussion – p12 – is there evidence of a subpopulation of AgRP neurons in which Trpv1 was more selectively expressed? What subpopulations are suggested herein? If included in Discussion, perhaps more providing more detail would be of benefit to readers.

14. Discussion – p12 – I don’t think that “feeding/refeeding” or “fasting/refeeding” experiments were performed in the current study, in which rats were all in fed state, with or without presence of food for the duration of the experiment. Data from the current study cannot necessarily be extrapolated. Thus, this statement should be revised.

15. Discussion – p13 – With DIO-induced AgRP neuron activity, is the propagation of adiposity also due to lethargy and decreased energy expenditure, rather than direct changes with WAT?

16. Figure 1 figure caption – please change “diving” to “dividing”.

17. Figure 2 – y-axes for right-side panels for 2b, 2c, and 2d should be indicated as “mean RER after gavage” or something to that effect

Reviewer #2 (Remarks to the Author):

This manuscript evaluates a novel role of AgRP neurons in mediating substrate utilization, independent of energy intake. The authors report that activation of AgRP neurons can rapidly alter body substrate utilization, with increased carbohydrate utilization and decreased fat utilization. The authors find the metabolic shifts were in accordance with increased lipogenesis and were blunted by FAS inhibitor. The study expanded the investigations of AgRP neurons in metabolism, in addition to appetite and energy intake. While this manuscript describes an interesting set of results which are novel and clearly described, I have significant issues and doubts about the data and manuscript in its current form.

- 1) The study focus on the whole-body substrate utilization. I wonder if RER (V_{CO2}/V_{O2}) itself is sufficient to represent the level of substrate utilization? Other evidence or reference would be helpful regarding this concern.
- 2) The levels of AgRP activation relative to control should be demonstrated (both in *AgrpTrpv1* mice and in *AgrphM3Dq* mice).
- 3) In figure 2b, glucose gavage could control substrate utilization towards carbohydrate, just like *agrp* neuron activation. Is there any change in glucose metabolism (for example, gluconeogenesis and glycogenolysis) in this case?
- 4) If activation of AgRP promotes glucose production and lipogenesis (figure 3), is the shift (from lipid to carbohydrate utilization) possibly caused by the increased glucose production?
- 5) In figure 3d, the genes of lipogenesis (*fasn*, *scd1*, *srebp1*) were unaltered, how to make the conclusion that AgRP neurons promote lipogenesis? Other evidence provided would be recommended.
- 6) As activation of AgRP neurons could alter the substrate utilization, how about the effect of inhibition of AgRP neurons using AAV-hM4D (Gi) in chow or obesogenic conditions?
- 7) The mechanisms how AgRP neurons convey to the peripheral to control substrate utilization need more investigations.

Reviewer #3 (Remarks to the Author):

The manuscript by Cavalcanti-de-Albuquerque and colleagues examines the role of hypothalamic AgRP neurons on whole-body substrate utilization. They report that activation of AgRP neurons is sufficient to increase carbohydrate oxidation and reduce fatty acid oxidation, independent of effects on feeding. Moreover, these effects were blocked with systemic administration of the fatty acid synthetase (FAS) inhibitor, C75. The authors also suggest that activity of AgRP neurons improved metabolic efficiency, leading to increased weight gain and adiposity, independent of changes in energy intake. Overall, the data generated are of interest, but some additional support is required to support the hypothesis.

It is already well-established that respiratory quotient is not only influenced by the composition, but also the quantity of the food consumed. While the authors acknowledge this, the data presented in Figures 1 and 2 are therefore somewhat confounded. Moreover, some of the strongest and most important data demonstrating that activation of AgRP neurons increases carbohydrate oxidation, independent of changes in energy intake in chow-fed mice is presented in Supplemental Figure 3 and it is suggested that this is moved into the main manuscript.

A large body of evidence using either pharmacological, DREADD and/or optogenetic approaches suggests that activation of AgRP neurons is sufficient to reduce energy expenditure. This effect is proposed to be mediated, in part, via reduced SNS outflow to BAT. Can the authors explain why in this model, activation of AgRP neurons failed to have any effect on energy expenditure?

The observation that the effect of AgRP activation on substrate utilization (reduced fatty acid oxidation and increased carbohydrate oxidation) is blocked by administration of the FAS inhibitor is compelling. However, there is potential concern, based on previous reports (MD Lane), that systemic administration of C75 may also have effects on hypothalamic NPY/AgRP and Pomc mRNA levels.

The authors suggest that activation of AgRP neurons shifts metabolism towards lipogenesis and promotes weight gain. The major conundrum the authors are required to address is that AgRP neurons are activated during fasting, an effect that not only drives hyperphagia, but reduces EE, and is a condition associated with increased mobilization of fuels (i.e. lipolysis, rather than lipogenesis). The Discussion that not all AgRP neurons are homogenous does not address this discrepancy since the approach does not distinguish between discrete populations of AgRP neurons, but activates them all.

For example, does activation of AgRP neurons also increase carbohydrate oxidation in fasted animals?

The studies and interpretation of data that activation of AgRP neurons favors weight gain in obesogenic conditions by regulating lipogenesis requires additional support. They report that AgRP neuronal activation induces greater weight gain in mice, independent of changes in energy intake. Is there an effect on body fat mass? Are these effects associated with expected biochemical changes in adipose tissue? To support the overarching hypothesis, the authors are required to demonstrate that activation of AgRP neurons reduces fatty acid oxidation and increases carbohydrate oxidation in HFD-fed mice, in the absence of food. Moreover, they are required to demonstrate that this increase in body weight is not attributable to an impaired ability to increase EE in this setting of hyperphagia. This is important considering that mice deficient in melanocortin signaling (Mc4r KO, Mc3r KO) exhibit an increased susceptibility to DIO, an effect not only due to hyperphagia, and nutrient partitioning, but reduced EE. The role of melanocortin receptors in mediating these effects also warrants attention in the Discussion.

There is some concern with the study design to address the question in Figure 4 given that HFD feeding by itself increases AgRP neuronal activity. Moreover, rather than simply focusing on whether activation of AgRP neurons is sufficient to exhibit effects, the more important physiological question is whether inhibition of AgRP neurons attenuates HFD-induced weight gain by regulating substrate utilization, independent of changes in energy intake.

There are several typos throughout the manuscript that warrant attention.

Reviewer #1 (Remarks to the Author):

This manuscript by Dr. Dietrich and colleagues reports novel observations that transient, chemogenetic activation of AgRP neurons rapidly and acutely alters whole body nutrient utilization measured using indirect calorimetry, independent of changes in energy expenditure. In fed mice with food withheld during the experiment, some changes in lipogenic/lipolytic genes/protein were observed in adipose tissue suggesting increased lipogenesis with acute AgRP neuron activation. AgRP neuron activation also increased body weight gain and modestly increased fat mass gain in mice given a controlled high fat feeding regime for 5 days. AgRP neuron activation was induced via ip capsaicin injection in mice that express capsaicin-sensitive channel Trpv1 in AgRP neurons, a mouse model the authors previously characterized (Dietrich et al, Cell 2015; Ruan et al, Cell 2014). This work extends knowledge from other studies that demonstrate that AgRP neuron activation can affect metabolism beyond its classical role on feeding behaviour.

The manuscript is well written with a coherent progression of ideas. The authors are commended for the work performed. The impact of this manuscript is limited by the following concerns:

RESPONSE. We thank the reviewer for the positive view on the manuscript. We also thank the reviewer for the all the constructive comments and we have now addressed them with new experimental data.

1. A widely known function of activated hypothalamic AgRP neurons to stimulate feeding behaviour in both fasted and sated states. Interestingly, however, one of the main claims is that AgRP neuron activation alters nutrient utilization “independent of food ingestion” (Introduction, p3, last line), or variations of the same [e.g. “absence of ingestive behaviours (Abstract), etc.] The authors have provided references to other studies in which AgRP neuron activation has metabolic effects beyond feeding, but I am not convinced that the studies herein fully exclude alterations in feeding behaviour induced by AgRP neuron activation. Fig 2 shows an effect of AgRP neuron activation on changing substrate utilization with and without glucose gavage in fed mice (presumably with no access to food; not clear in Methods), and the mean effect being greater in AgRP-Trpv1 mice with glucose gavage than control counterparts.

RESPONSE.

We have included a more detailed explanation of the tests used and clarified the language throughout the text. We have also clarified the experiments described in Figure 2. Mice did not have access to food and activation of Agrp neurons increased RER independent of glucose gavage. When Agrp neurons were activated, the increase in RER (and other metabolic parameters) was similar in magnitude in mice that received gavage with vehicle or glucose.

However, could that moderate increase be due in part to increased food seeking activity of the mice? In the absence of food/glucose, fed mice with AgRP neuron activation show increased activity (Suppl Fig 3f), possibly Agrp-stimulated food seeking behaviour, and as the authors also state, physical activity can increase RER. Interestingly, this increase in activity coincides at ~30 min (Fig 3f), the time in which the added effect of AgRP activation is seen in Fig 2.

RESPONSE.

The increase in RER upon activation of Agrp neurons is very robust and cannot be mimicked even with the highest dose of glucose delivered by gavage. During exercise RER increases - to these high levels – when above the aerobic threshold, which occurs during exhaustive forced exercise and not during home cage activity levels. While we did not collect data for activity during the glucose gavage experiment (there was a hardware problem), we did collect activity levels in other experiments in which we activated Agrp neurons in the absence of food. We now present a more detailed analysis of the changes in activity and RER. We have moved the supplementary figure to main figure (**Fig. 3**) to emphasize the changes in activity levels and the uncoupling between activity and RER. We used linear regression analysis (**Fig. 3f-g**) and found that activity levels and RER were weakly correlated in both control and *Agrp^{Trpv1}* mice as indicated by the slopes being significantly higher than zero (control: $r^2 = 0.244$, $F_{1,63} = 20.12$, $P < 10^{-4}$; *Agrp^{Trpv1}*: $r^2 = 0.155$, $F_{1,63} = 11.21$, $P < 10^{-3}$). However, the slopes were not statistically different from each other (control: $slope = 1.68 \times 10^{-4} \pm 0.37 \times 10^{-4}$; *Agrp^{Trpv1}*: $slope = 1.21 \times 10^{-4} \pm 0.36 \times 10^{-4}$; $F_{1,126} = 0.82$, $P = 0.36$), suggesting the physiological relationship between RER and activity levels is not altered by Agrp neuron activation. Importantly, across different activity levels activation of Agrp neurons increased RER, as evidenced by the parallel linear regression lines and by statistically different intercepts (control: $intercept = 0.766 \pm 0.005$; *Agrp^{Trpv1}*: $intercept = 0.878 \pm 0.007$, $F_{1,127} = 201.5$, $P < 10^{-4}$). Thus, the increase in RER upon Agrp neuron activation is unlikely to be due to activity levels.

In Figure 1 in the presence of food, how much are the mice eating during the time course of the experiment? The increase in carbohydrate utilization in the presence of food appears much more prolonged (sustained for few hours) compared to experiments in the absence of food, in which carbohydrate utilization peaks at t=30 mins. Food intake data for Fig 1 experiments would also be useful.

RESPONSE.

We now provide food intake data for Figure 1 and Supplementary Figure 1. In the presence of ingestion, the increase in RER is more sustained likely due to the postprandial effects of carbohydrate consumption on de novo lipogenesis.

2. Results – Figure 1 – Authors state that RER is calculated from dividing VCO₂ by VO₂ (Methods, p16). If there is no difference between AgRP-TRPV1 and control mice in VCO₂ (Fig 1c) or VO₂ (Fig 1d) why is RER so drastically different between the 2 groups?

RESPONSE.

We now provide inserts in the figure to illustrate the (non-statistically significant) changes in VO₂ and VCO₂, which account for the change in RER (**Fig. 1f-g**). The reviewer should note that VO₂ and VCO₂ are absolute values with a much wider range, while RER is a fraction with a much narrower range. Small changes in VO₂ and VCO₂ signify substantial shifts in substrate utilization, which is accounted for when RER is calculated.

3. Blood glucose levels are shown in Fig 3C – could blood glucose data also be provided in glucose gavage experiments (Fig 2 and Suppl Fig 2)? Blood glucose profiles could be useful in discussing nutrient utilization results.

RESPONSE.

Recording circulating factors while animals are tested in metabolic cages would require opening the cages repeatedly and subjecting mice to multiple bleeding. Unfortunately, this procedure is difficult to perform in metabolic chambers and not allowed in our Institution by the IACUC Committee. Thus, we did not collect blood glucose/NEFA data during this experiment. However, the dose of glucose used was similar to typical GTT/oGTT tests. Thus, glucose levels increase in a similar rate as the RER. Importantly, delivery of glucose did not change the metabolic shift that occurred upon activation of Agrp neuron compared to mice that did not receive glucose (see response to criticism #1 above). Also, activation of Agrp neurons did not change circulating glucose levels (Fig. 4c). These results suggest the shift in metabolism is not due to an increase in glucose release by the liver, kidney, or intestines. Future studies should address whether Agrp neurons directly communicate with these tissues to alter glucose metabolism.

4. In Ref 24, one of the metabolic effects of activation of AgRP neurons was inhibition sympathetic nerve activity (SNA), at least to brown adipose. Inhibition of SNA would also decrease lipolysis, which would also lead to decreased blood NEFA levels. Inhibition of SNA could also decrease HSL activity, which agrees with HSL western data (Fig 3e). The involvement of AgRP neuron activation and the sympathetic nervous system could be touched on in Discussion.

RESPONSE.

We have included discussion on the role of the SNA in mediating the peripheral effects of Agrp neurons. We also provide new experimental data showing that a pharmacological agonist of adrenergic beta3-receptors abrogated the effects of Agrp neurons on metabolic switch (Fig. 5), in line with Agrp neuron activation decreasing the activity of the SNA.

5. The changes induced by AgRP neuron activation on pHSL protein levels and gene expression of hk2 and ppar-alpha in WAT as indicators of promoting lipogenesis is strengthened by experiments using FAS inhibitor C75. However, a stronger piece of evidence is simply fat mass. In normal chow fed mice (e.g. not HFD data in Fig 4), is there an increase in adiposity, particularly given the lasting

effects of AgRP stimulation as shown in Fig 1? Lipogenesis would be expected to be lower in mice during the light cycle, during which AgRP neurons were activated, thus may help to amplify changes in lipogenesis.

RESPONSE.

It is technically difficult to see acute changes in fat mass due to the resolution of the techniques used to measure fat mass. In our hands, in vivo measurements such as MRI do not have enough resolution to detect changes in fat mass in the order of milligrams. To overcome this limitation, we daily activated *Agrp* neurons and measured changes in body mass and fat mass over a period of 10 days. We now show in Supplementary Figure 4 that activation of *Agrp* neurons increased fat deposition and metabolic efficiency, supporting that the increase in food intake is not sufficient to explain the changes in the accumulation of fat mass and body weight.

We also include new loss-of-function experiments. We studied control and *Agrp*^{DTR} mice, which express the receptor for diphtheria toxin (DT) exclusively in *Agrp* neurons. By injecting DT peripherally, *Agrp* neurons are ablated in a time-controlled manner. In adults, ablation of *Agrp* neurons leads to aphagia and death (Luquet et al. Science 2005). Thus, we had to develop a different feeding schedule to bypass the aphagia after *Agrp* neuron ablation. We devised an enteral feeding scheme using a gastrostomy tube to deliver controlled amounts of liquid diet into the stomach of mice (**Fig. 7a**). After tube implantation, mice were allowed to recover from surgery and acclimate to liquid diet. We then gradually increased the amount of diet infused in the stomach, until we reached positive energy balance (animals started to gain body weight). In this feeding schedule, ablation of *Agrp* neurons did not lead to death or weight loss (**Fig. 7b**). Both control and *Agrp*^{DTR} animals continue to increase body weight (**Fig. 7b**). Ablation of *Agrp* neurons led to a significant decrease in fat mass gain (**Fig. 7c**) but not lean mass (delta lean mass: control = -0.07 ± 0.30 , *Agrp*^{DTR} = 0.13 ± 0.49 ; $U = 9$, $P = 0.45$, Mann-Whitney Test) as measured by repeated MRI scans before the first DT injection and at the end of the study. Six days after the first injection of DT, we dissected the fat tissue for biochemical analysis and the brain to confirm ablation of *Agrp* neurons. Only minimal residual *Agrp* fibers were visualized in the arcuate nucleus (**Fig. 7d**), confirming the ablation of *Agrp* neurons. In the white adipose tissue, we measure genes involved in lipid metabolism and found decreased expression of several genes involved in lipogenesis (**Fig. 7e**).

6. Experiments in Fig 4 are nicely designed, but a few concerns arise: (a) HFD was given for 5 days, but body weight data for the 5th day (Fig 4i) is missing. Please include day 5 data. (b) What is the axis showing in Fig 4i – is it BW fold increase? Please clarify. (c) How many days following HFD was data for Fig 4j and 4k obtained? (d) In this step-wise increase of HFD feeding for 5 days, do mice become more “obese”, compared to regular show fed mice, even without chemogenetic activation of AgRP neurons? If yes, then would AgRP neurons be intrinsically activated (by DIO, as previously shown), and what then is the added stimulation of AgRP neurons? For example, in this DIO model, capsaicin is still injected daily to activate AgRP. (e) Fig 4k represents very modest changes

differences in fat mass gain between the 2 groups. Is this difference in fat mass gain sufficient to explain the difference in body weight changes?

RESPONSE.

We thank the reviewer for raising these points. (a) We fixed the missing point in Figure 4 (now 6i) and the corresponding statistical data. (b) We have clarified the Y axis label – we now report delta body weight in grams relative to day 0. (c) At day 6 – basically after 5 full days on high-fat diet. We clarified this point in the text and methods. (d) Measuring the activity of Agrp neurons directly in these long-term scenarios is technically challenging. To help address this concern, we have included loss-of-function data (Figure 7 and previous response) that supports the role of Agrp neuron activity during positive energy balance in the accumulation of fat mass. (e) Changes in body composition were measured using MRI. While the changes in fat mass were not of the same magnitude that changes in body mass, they were not concomitant with changes in lean mass (data now provided in the text). It is possible that other sources of changes in body composition, such as water content, are altered in this experimental condition. We have now added a second experimental condition activating Agrp neurons with hM3Dq, showing larger increase in metabolic efficiency and fat accumulation (Supplementary Figure 5).

7. The title of the manuscript and elsewhere in the text suggests that the hypothalamus is the specific brain region involved in the metabolic effects demonstrated. Whereas AgRP neurons are well known to be found in specific brain regions to control energy metabolism (e.g. ARC), do the authors have evidence in their model that only the hypothalamus is affected?

RESPONSE.

We have addressed this concern by decreasing the emphasis in the hypothalamus as the sole mediator of these metabolic changes, as our data do not support this conclusion. We have also discussed possible circuits involved in the control of lipogenesis and RER.

8. Suppl Fig 4 – AAV-DIO-hM3D(Gq)mCherry mice are not found in the Methods. Please include more details. Additionally, have these mice been validated elsewhere? This figure is missing control mice – what is the effect of CNO alone, and why is CNO action on its receptor hM3D(q) in AgRP neurons indicative of neuronal activation? Are HM3D ubiquitously expressed? Readers would benefit from additional information about this model if used in the present study.

RESPONSE.

We now include a description of this model in the methods. Many laboratories have this model to study the function of Agrp neurons and we now cite these previous references.

Minor Considerations

9. Methods – p16 – Formulas for fat utilization and carbohydrate utilization are

missing units. Please provide units for the constants used in the formula.

RESPONSE. We now include the modifications as suggested.

10. Methods – p15 – In the subsection, “Drugs”, please provide references for the reported studies indicated in the text.

RESPONSE. We have added the modifications as suggested.

11. Discussion – p11 – “elevated activity of AgRP neurons” should be changed to “activation of AgRP neurons”.

RESPONSE. We now include the modification as suggested.

12. Discussion – 11 – ref 43 is a submitted work and not published and should perhaps then be noted as “unpublished data” unless status of submission is changed.

RESPONSE. We have removed this reference as the manuscript is still pending publication and the reference is not critical for the argument.

13. Discussion – p12 – is there evidence of a subpopulation of AgRP neurons in which Trpv1 was more selectively expressed? What subpopulations are suggested herein? If included in Discussion, perhaps more providing more detail would be of benefit to readers.

RESPONSE. As we have reported in Dietrich et al. Cell 2015, the activation of Agrp neurons in Agrp-Trpv1 mice is not selective to an anatomically specified Agrp neuron population. We did not investigate specific subpopulations of Agrp neurons driving these metabolic changes. In line with the criticism of the reviewer, we removed this paragraph of the discussion as it was speculative only.

14. Discussion – p12 – I don’t think that “feeding/refeeding” or “fasting/refeeding” experiments were performed in the current study, in which rats were all in fed state, with or without presence of food for the duration of the experiment. Data from the current study cannot necessarily be extrapolated. Thus, this statement should be revised.

RESPONSE. We have removed this paragraph of the discussion, in line with the response for the previous criticism.

15. Discussion – p13 – With DIO-induced AgRP neuron activity, is the propagation of adiposity also due to lethargy and decreased energy expenditure, rather than direct changes with WAT?

RESPONSE. In our laboratory, we have never observed lethargy during diet-induced obesity. We now provide new data showing the involvement of the SNS in the rapid effects of Agrp neurons on peripheral metabolism and loss-of-function experiments showing ablation of Agrp neurons decreased the gain in fat mass during positive energy balance. It was not possible to perform the loss-of-function experiments inside metabolic cages as the animals had to be connected to an enteral feeding tube. The

rapid changes in peripheral metabolism and WAT biochemistry strongly suggest Agrp neurons directly control this tissue. The control of WAT function by Agrp neurons does not exclude a contribution of Agrp neurons in energy expenditure and other metabolic changes that can occur concomitantly or in a more delayed manner.

16. Figure 1 figure caption – please change “diving” to “dividing”.

RESPONSE. We have fixed this typo.

17. Figure 2 – y-axes for right-side panels for 2b, 2c, and 2d should be indicated as “mean RER after gavage” or something to that effect

RESPONSE. We have re-labeled the axes as suggested.

Reviewer #2 (Remarks to the Author):

This manuscript evaluates a novel role of AgRP neurons in mediating substrate utilization, independent of energy intake. The authors report that activation of AgRP neurons can rapidly alter body substrate utilization, with increased carbohydrate utilization and decreased fat utilization. The authors find the metabolic shifts were in accordance with increased lipogenesis and were blunted by FAS inhibitor. The study expanded the investigations of AgRP neurons in metabolism, in addition to appetite and energy intake. While this manuscript describes an interesting set of results which are novel and clearly described, I have significant issues and doubts about the data and manuscript in its current form.

1) The study focus on the whole-body substrate utilization. I wonder if RER (VCO_2/VO_2) itself is sufficient to represent the level of substrate utilization? Other evidence or reference would be helpful regarding this concern.

RESPONSE.

Measuring the RER allows to differentiate lipid catabolism (RER ~0.7) from carbohydrate catabolism (RER ~1.0). Intermediate values reflecting a mix of lipid and carbohydrate catabolism. RER is used in indirect calorimetry by measuring respired gases (VCO_2 and VO_2). In animals, the measurement of gases estimates the respiratory quotient (RQ), which can only be measured precisely at the cellular level. Therefore, RER represents the sum of all oxidative processes happening in the organism, reason why is used as a valid tool to calculate substrate utilization. Other techniques, including tracing, can indicate specific metabolic routes, but cannot estimate overall metabolism and have much slower time resolution. Thus, in living organisms RER calculation is the best estimate one can have of substrate utilization. It can be interpreted based on Hess's Law of Constant Heat Summation, in which the enthalpy change in a chemical reaction is independent of the number of reaction steps. Thus, oxidizing food in a bomb calorimeter or in an animal yields equivalent energy. If one measures the RER for long enough, it should equal the RQ calculated for the food ingested in an animal that is not changing body mass. It is important to note that in stress circumstances, as for example during strenuous exercise above the VO_{2max} , the rate of oxygen supply is no longer sufficient to maintain aerobic metabolism and the muscles start to respire anaerobically. In this case, lactate begins to accumulate driving a drop in blood pH, and a shift in buffering by bicarbonate/ CO_2 , leading to the generation of extra CO_2 , which is exhaled and detected by indirect calorimetry. In other words, during anaerobic respiration O_2 is not consumed and CO_2 is produced, therefore RER is infinite. Indeed, the shift towards anaerobic metabolism during exercise is detected by a sharp increase in RER, indicating the breakpoint (or VO_{2max}). Animals do not voluntarily exercise to the level of VO_{2max} ; it can only be induced by forced exercise. We now provide references in the methods section to support the use of RER to estimate whole-body nutrient utilization.

2) The levels of AgRP activation relative to control should be demonstrated (both in *AgRP^{Trpv1}* mice and in *AgRP^{M3Dq}* mice).

RESPONSE.

We have characterized the *AgRP-^{Trpv1}* animals in our previous publications (Dietrich et al., Cell 2015) and we now cite this manuscript in the text. Using tools such as optogenetics, TRPV1 or

hM3Dq, it is possible to activate the vast majority of neurons in the arcuate nucleus (~90% in our animals as measured by c-fos). Thus, in our studies, we activated the vast majority of Agrp neurons. Unfortunately, the in vivo dynamics of Agrp neurons in response to these cell actuators is not known. The lab of Mark Anderman is the only one to have successfully recorded Agrp neuronal activity in vivo using electrophysiology. However, in their studies, they used head-fixed mice making it difficult to extrapolate their findings to freely moving animals in more physiological contexts. We have also added new loss-of-function experiments (Fig. 7), in which Agrp neurons are ablated. These ablation experiments show that during positive energy balance animals without Agrp neurons accumulate less fat mass.

3) In figure 2b, glucose gavage could control substrate utilization towards carbohydrate, just like agrp neuron activation. Is there any change in glucose metabolism (for example, gluconeogenesis and glycogenolysis) in this case?

RESPONSE.

Activation of Agrp neurons increased substrate utilization to higher levels and in a more sustained manner compared to the highest dose of glucose. Also, activation of Agrp neurons even without glucose gavage altered substrate utilization. Other laboratories have investigated in more detail the participation of Agrp neurons in hepatic glucose metabolism. The laboratory of Jens Bruning (Steculorum et al., Cell 2016) did not find differences in hepatic glucose production at baseline or in insulin-suppressed rate upon acute activation of Agrp neurons. Therefore, these results suggest Agrp neurons control lipogenesis in the fat independently of glucose production by the liver.

4) If activation of AgRP promotes glucose production and lipogenesis (figure 3), is the shift (from lipid to carbohydrate utilization) possibly caused by the increased glucose production?

RESPONSE.

Our results do not support the hypothesis that Agrp neurons promote glucose production to increase lipogenesis (see response #3).

5) In figure 3d, the genes of lipogenesis (fasn, scd1, srebp1) were unaltered, how to make the conclusion that AgRP neurons promote lipogenesis? Other evidence provided would be recommended.

RESPONSE.

Nutrients and hormones classically regulate these genes during post-prandial conditions during an extended period. As we analyzed changes in gene expression shortly after Agrp neuron activation, it is expected that changes in genes with low kinetics to be not altered. To overcome this limitation, we measured changes in HSL phosphorylation, which is a more direct evidence indicating increases in lipogenesis upon Agrp neuron activation. These results are reinforced by the blockade of FAS by C75, which acutely reverted the effects of Agrp neuron activation on RER. Additional evidence supporting the role of Agrp neurons in lipogenesis is the increased expression of isoform II of hexokinase in the fat. This isoform is associated with increases in glucose flux and, consequently, lipogenesis.

We also include new loss-of-function experiments. We studied control and *Agrp*^{DTR} mice, which express the receptor for diphtheria toxin (DT) exclusively in *Agrp* neurons. By injecting DT peripherally, *Agrp* neurons are ablated in a time-controlled manner. In adults, ablation of *Agrp* neurons leads to aphagia and death (Luquet et al. Science 2005). Thus, we had to develop a different feeding schedule to bypass the aphagia after *Agrp* neuron ablation (see Figure 6). We devised an enteral feeding scheme using a gastrostomy tube to deliver controlled amounts of liquid diet into the stomach of mice. After tube implantation, mice were allowed to recover from surgery and acclimate to liquid diet. We then gradually increased the amount of diet infused in the stomach, until we reached positive energy balance (animals started to gain body weight). In this feeding schedule, ablation of *Agrp* neurons did not lead to death or weight loss. Both control and *Agrp*^{DTR} animals continue to increase body weight. Ablation of *Agrp* neurons led to a significant decrease in fat mass gain but not lean mass (delta lean mass: control = -0.07 ± 0.30 , *Agrp*^{DTR} = 0.13 ± 0.49 ; $U = 9$, $P = 0.45$, Mann-Whitney Test) as measured by repeated MRI scans before the first DT injection and at the end of the study. Six days after the first injection of DT, we dissected the fat tissue for biochemical analysis. In the white adipose tissue, we observe a decrease in *fasn*, *scd1* and *srebp1* in the WAT of animals in which *Agrp* neurons were ablated during positive energy balance (Fig. 7e).

6) As activation of AgRP neurons could alter the substrate utilization, how about the effect of inhibition of AgRP neurons using AAV-hM4D (Gi) in chow or obesogenic conditions?

RESPONSE.

Loss of function experiments are notoriously harder to perform than gain of function. In the case of *Agrp* neurons, acute neuronal inhibition suppresses appetite, making it difficult for us to study change in fat gain over prolonged periods of time. We have chosen a very efficient method to study the loss of function of *Agrp* neurons, which is neuronal ablation using diphtheria toxin receptor (Luquet et al. Science 2005). Using the *Agrp*-DTR knock-in line generated by the lab of Richard Palmiter, it is possible to ablate virtually the entire population of *Agrp* neurons acutely (within few days) in the adult mouse. Acute ablation of *Agrp* neurons leads to aphagia and death. To overcome these limitations, we have performed intragastric infusions of diet in control and *Agrp*-DTR mice (see response to criticism #5). Upon ablation of *Agrp* neurons, we 'clamped' food infusion to a positive energy balance to induce weight gain. Intragastric infusion of food prevented death in *Agrp* neuron ablated mice. In support of our results on the role of *Agrp* neurons in lipogenesis, ablation of these neurons led to decreased fat gain under intragastric dieting. These results are reported in Figure 7.

7) The mechanisms how AgRP neurons convey to the peripheral to control substrate utilization need more investigations.

RESPONSE.

A plausible explanation for the peripheral effects of *Agrp* neurons on substrate utilization is a decrease in SNS activity. We have now provided novel data showing that the use of a pharmacological agonist of beta3-receptors (adrenergic receptors specific for the fat) acutely blocks the effects of *Agrp* neurons on RER but not on food intake (Fig. 5). We also discussed this point in the text.

Reviewer #3 (Remarks to the Author):

The manuscript by Cavalcanti-de-Albuquerque and colleagues examines the role of hypothalamic AgRP neurons on whole-body substrate utilization. They report that activation of AgRP neurons is sufficient to increase carbohydrate oxidation and reduce fatty acid oxidation, independent of effects on feeding. Moreover, these effects were blocked with systemic administration of the fatty acid synthetase (FAS) inhibitor, C75. The authors also suggest that activity of AgRP neurons improved metabolic efficiency, leading to increased weight gain and adiposity, independent of changes in energy intake. Overall, the data generated are of interest, but some additional support is required to support the hypothesis.

It is already well-established that respiratory quotient is not only influenced by the composition, but also the quantity of the food consumed. While the authors acknowledge this, the data presented in Figures 1 and 2 are therefore somewhat confounded. Moreover, some of the strongest and most important data demonstrating that activation of AgRP neurons increases carbohydrate oxidation, independent of changes in energy intake in chow-fed mice is presented in Supplemental Figure 3 and it is suggested that this is moved into the main manuscript.

RESPONSE.

We thank the review for the commentaries and for the suggestion to move Supplemental Figure 3 into the main manuscript (now as Figure 3). Results in figures 1 and 2 result from completely different experimental paradigms. While in Figure 1 animals had access to ad libitum food, in Figure 2 they received a gavage infusion of glucose (or vehicle). In Figure 1, activation of *Agrp* neurons led to increase food intake, which per se could explain the increase in RER. We now reported the changes in feeding in Figure 1 and Supplemental Figure 1. Experiment reported in Figure 2 aims to resolve this question by infusing the same amount of glucose in control and *Agrp* neuron activated mice. It was remarkable to find that injection (or not) of glucose do not alter the increase in RER promoted by *Agrp* neuron activation. This increase was also of larger magnitude than oral glucose alone (Figure 2).

A large body of evidence using either pharmacological, DREADD and/or optogenetic approaches suggests that activation of AgRP neurons is sufficient to reduce energy expenditure. This effect is proposed to be mediated, in part, via reduced SNS outflow to BAT. Can the authors explain why in this model, activation of AgRP neurons failed to have any effect on energy expenditure?

RESPONSE.

We are not aware of any publication using optogenetics to measure energy expenditure upon *Agrp* neuron activation. Steculorum et al. (Cell 2016) used optogenetics and DREADDs to study glucose metabolism and BAT function upon activation of *Agrp* neurons, but they did not measure energy expenditure. In Krashes et al. (JCI 2010) the authors reported a drop in VO₂ but did not report energy expenditure upon *Agrp* neuron

activation in the presence of food using hM3Dq. In our experiments using both Trpv1 and hM3Dq, we did not find similar results in the presence or absence of food. While we do not know exactly the origins of these discrepancies, a recent report from Burke et al. (eLife 2017) indicate that small differences in experimental conditions have a large impact on the effects of Agrp neuron activation on energy expenditure and iBAT thermogenesis. For example, when they activated Agrp neurons in the presence of food or in the presence of sensory detection of food (caged food, not available for food intake), no changes in energy expenditure were detected. When using new clean cages, they found robust changes in energy expenditure. In the studies reported here, all animals were acclimated to the cages for several days before the experiments (for example, Figure S1). Thus, it is possible that some subtle change in the experimental condition blocked the effects of Agrp neuron activation on energy expenditure. We have now included in the discussion the possibility that the effects we observed in our study are related to the anticipatory changes in lipogenesis that occur before feeding just before that dark cycle in mice.

The observation that the effect of AgRP activation on substrate utilization (reduced fatty acid oxidation and increased carbohydrate oxidation) is blocked by administration of the FAS inhibitor is compelling. However, there is potential concern, based on previous reports (MD Lane), that systemic administration of C75 may also have effects on hypothalamic NPY/AgRP and Pomc mRNA levels.

RESPONSE.

We agree with the reviewer that systemic administration of C75 may have other effects. We were careful to choose a low dose of C75 (10 mg/kg) to prevent side effects detected with higher doses. Also, the effects of C75 on hypothalamic circuits and feeding have been reported for longer time points (24h). In our studies, we only analyzed the first hour after injection of C75 to prevent the strong metabolic effects secondary to inhibition of FAS that occurs after treatment with C75. Supporting our approach, treatment of control mice with C75 did not change RER or fat/carbohydrate metabolism during the first hour post-treatment (Figure 4). We have also added new data using a pharmacological agonist of beta3-adrenergic receptors that also blocked the effect of Agrp neuron activation on RER (Figure 5), supporting the mechanistic explanation that Agrp neurons control FAS activity in the white fat via the SNS. Finally, we also added new loss-of-function experiments showing ablation of Agrp neurons decreases the gain in fat mass during positive energy balance (Figure 7). These added results argue against an indirect effect of C75 on metabolism as an explanation for our findings.

The authors suggest that activation of AgRP neurons shifts metabolism towards lipogenesis and promotes weight gain. The major conundrum the authors are required to address is that AgRP neurons are activated during fasting, an effect that not only drives hyperphagia, but reduces EE, and is a condition associated with increased mobilization of fuels (i.e. lipolysis, rather than lipogenesis). The Discussion that not all AgRP neurons are homogenous does not address this

discrepancy since the approach does not distinguish between discrete populations of AgRP neurons, but activates them all.

For example, does activation of AgRP neurons also increase carbohydrate oxidation in fasted animals?

RESPONSE.

Agrp neurons have increased activity during food deprivation (Grove et al., 2003; Hahn et al., 1998; Takahashi and Cone, 2005) but also during high-fat feeding as recorded using slice physiology (Baver et al., 2014; Diano et al., 2011; Dietrich et al., 2013b; Wei et al., 2015). We have previously demonstrated (Dietrich et al., 2013a) that mice in which the Agrp neurons had impaired cellular activation during high-fat feeding were protected against adiposity. We have now performed loss-of-function experiments, ablating Agrp neurons during positive energy balance. We show (Fig. 7) ablation of Agrp neurons decrease the gain in fat mass and the expression of lipogenesis-related genes in the WAT. We now discuss our data on the relevance of Agrp neurons to fat accumulation during positive energy balance in line with these findings.

The studies and interpretation of data that activation of AgRP neurons favors weight gain in obesogenic conditions by regulating lipogenesis requires additional support. They report that AgRP neuronal activation induces greater weight gain in mice, independent of changes in energy intake. Is there an effect on body fat mass? Are these effects associated with expected biochemical changes in adipose tissue? To support the overarching hypothesis, the authors are required to demonstrate that activation of AgRP neurons reduces fatty acid oxidation and increases carbohydrate oxidation in HFD-fed mice, in the absence of food. Moreover, they are required to demonstrate that this increase in body weight is not attributable to an impaired ability to increase EE in this setting of hyperphagia. This is important considering that mice deficient in melanocortin signaling (Mc4r KO, Mc3r KO) exhibit an increased susceptibility to DIO, an effect not only due to hyperphagia, and nutrient partitioning, but reduced EE. The role of melanocortin receptors in mediating these effects also warrants attention in the Discussion.

RESPONSE.

We do not claim our results are the only regulatory mechanism by which Agrp neurons control metabolism. They are in addition to other well-known functions of Agrp neurons in feeding and thermogenesis. In the overnight experiments, the resolution of MRI is not sufficient to find small changes in fat mass. In longer experiments, we do show an increase in fat mass (Figures 6, 7 and S4, S5 – new data). We also provide new experiments with ablation of Agrp neurons, showing a decrease in the gain of fat mass (Figure 7). Using loss-of-function (Fig. 7), we found a decrease in the expression of lipogenesis-related genes in the WAT. We have discussed the potential role of melanocortin signaling as suggested by the reviewer.

There is some concern with the study design to address the question in Figure 4 give that HFD feeding by itself increases AgRP neuronal activity. Moreover, rather than simply focusing on whether activation of AgRP neurons is sufficient to exhibit effects, the more important physiological question is whether inhibition of AgRP neurons attenuates HFD-induced weight gain by regulating substrate utilization, independent of changes in energy intake.

RESPONSE.

In the case of Agrp neurons, acute neuronal inhibition suppresses appetite, making it difficult for us to study change in fat gain over prolonged periods of time. We have chosen a very efficient method to study the loss of function of Agrp neurons, which is neuronal ablation using diphtheria toxin receptor (Luquet et al. Science 2005). Using the Agrp-DTR knock-in line generated by the lab of Richard Palmiter, it is possible to ablate virtually the entire population of Agrp neurons acutely (within few days) in the adult mouse. Acute ablation of Agrp neurons leads to aphagia and death. To overcome these limitations, we have performed intragastric infusions of diet in control and Agrp-DTR mice. Upon ablation of Agrp neurons, we 'clamped' food infusion to a positive energy balance to induce weight gain. Intragastric infusion of food prevented death in Agrp neuron ablated mice. In support of our results on the role of Agrp neurons in lipogenesis, ablation of these neurons led to decreased fat gain under intragastric dieting. These results are reported in Figure 7.

There are several typos throughout the manuscript that warrant attention.

RESPONSE.

We have sent the manuscript to professional editorial editing to correct for typos.

References

- Baver, S.B., Hope, K., Guyot, S., Bjorbaek, C., Kaczorowski, C., and O'Connell, K.M. (2014). Leptin modulates the intrinsic excitability of AgRP/NPY neurons in the arcuate nucleus of the hypothalamus. *The Journal of neuroscience : the official journal of the Society for Neuroscience* *34*, 5486-5496.
- Diano, S., Liu, Z.W., Jeong, J.K., Dietrich, M.O., Ruan, H.B., Kim, E., Suyama, S., Kelly, K., Gyengesi, E., Arbiser, J.L., *et al.* (2011). Peroxisome proliferation-associated control of reactive oxygen species sets melanocortin tone and feeding in diet-induced obesity. *Nature medicine* *17*, 1121-U1130.
- Dietrich, M.O., Liu, Z.-W., and Horvath, T.L. (2013a). Mitochondrial Dynamics Controlled by Mitofusins Regulate *Agrp* Neuronal Activity and Diet-Induced Obesity. *Cell* *155*, 188-199.
- Dietrich, M.O., Liu, Z.W., and Horvath, T.L. (2013b). Mitochondrial dynamics controlled by mitofusins regulate *agrp* neuronal activity and diet-induced obesity. *Cell* *155*, 188-199.
- Grove, K.L., Chen, P., Koegler, F.H., Schiffmaker, A., Susan Smith, M., and Cameron, J.L. (2003). Fasting activates neuropeptide Y neurons in the arcuate nucleus and the paraventricular nucleus in the rhesus macaque. *Brain Res Mol Brain Res* *113*, 133-138.
- Hahn, T.M., Breininger, J.F., Baskin, D.G., and Schwartz, M.W. (1998). Coexpression of *Agrp* and NPY in fasting-activated hypothalamic neurons. *Nature neuroscience* *1*, 271-272.
- Takahashi, K.A., and Cone, R.D. (2005). Fasting induces a large, leptin-dependent increase in the intrinsic action potential frequency of orexigenic arcuate nucleus neuropeptide Y/Agouti-related protein neurons. *Endocrinology* *146*, 1043-1047.
- Wei, W., Pham, K., Gammons, J.W., Sutherland, D., Liu, Y., Smith, A., Kaczorowski, C.C., and O'Connell, K.M. (2015). Diet composition, not calorie intake, rapidly alters intrinsic excitability of hypothalamic AgRP/NPY neurons in mice. *Scientific reports* *5*, 16810.

Reviewers' comments:

Reviewer #1 (Remarks to the Author):

This revised manuscript by Dr. Dietrich and colleagues is improved and contains key revisions. The majority of my concerns have been adequately addressed. Importantly, AgRP neuron activation indeed shows a robust acute increase in food intake, and this is addressed now with inclusion of feeding data in Figure 1 and in the text and may explain the disparity in the sustained RER and substrate utilization effects observed in Figure 1 compared with the more short-lived effects of AgRP activation in subsequent figures.

Unfortunately, blood glucose data for the glucose gavage experiments (Figure 2) cannot be provided. The authors show that blood glucose levels are not appreciably different with AgRP activation in Figure 4c, but this was not in response to a glucose challenge. Measurements of glucose handling after a glucose challenge, including how quickly it is cleared, are different from measuring basal glucose blood levels. Perhaps a brief statement of this limitation could be included in the text. Technical reasons are cited in the response to reviewers.

The linear regression data between activity and RER is interesting, and it would certainly be anticipated that AgRP neuron activation in the absence of food would stimulate foraging behaviour. The wording on page 4 that describes the regression analysis could benefit from some re-wording (for example, what is meant by: (i) "anticipation in the effects of AgRP neuron activation"? (ii) "slopes been significantly higher than zero"?).

Figure 4 shows data to address the promotion of lipogenesis as an effect of AgRP activation. It can be noted that the difference in NEFA data (~ 1.6 vs 1.4 mEq/L) is quite small, even if statistically significant. To further implicate the promotion of lipogenesis as an effect of AgRP activation, the authors have now added in addition to their lipogenic data and C75 FAS inhibitor experiments a chronic study over 10 days to show that AgRP activation increases fat mass to an extent not explained by increased food intake alone. Additionally, new loss of AgRP function experiments show that under positive energy conditions in mice fed through a gastric tube, AgRP neuron ablation leads to decreased fat mass gain. Was PNPLA2 gene expression measured? In WAT, key mechanisms regulating lipolysis involve not only HSL but also ATGL and perilipin (the latter particularly if beta-adrenergic stimulation is involved).

In the AgRP ablation study, during positive energy balance, is it known to what level AgRP neurons in control mice become activated? This question would be similar also control mice in the high-fat diet study, given that high-fat feeding also increases AgRP neuron activity. Do they ever reach levels comparable to AgRP-activated (intact) mice?

Of note, the authors have now also included exciting new experimental findings, including important experiments that begin to reveal the role of beta-adrenergic suppression, which mediates some of the metabolic effects of AgRP neuron activation (Figure 5). The use of the beta-adrenergic receptor agonist appears to completely reverse the metabolic effects of AgRP activation. Interestingly, the use of the beta-adrenergic receptor agonist, CL 316243, has previously been shown to decrease RER and increase plasma FFA in control and non-obese T2D mice (PMID: 16682489), which differs somewhat from the lack of effect that this agonist has in control mice.

Reviewer #2 (Remarks to the Author):

In this revision, the authors have addressed my points both experimentally and through further

literature support. The new data have adequately addressed my concerns and recommend acceptance.

Reviewer #3 (Remarks to the Author):

The authors have addressed the Reviewers concerns in a satisfactory manner.

Reviewers' comments:

Reviewer #1 (Remarks to the Author):

This revised manuscript by Dr. Dietrich and colleagues is improved and contains key revisions. The majority of my concerns have been adequately addressed. Importantly, AgRP neuron activation indeed shows a robust acute increase in food intake, and this is addressed now with inclusion of feeding data in Figure 1 and in the text and may explain the disparity in the sustained RER and substrate utilization effects observed in Figure 1 compared with the more short-lived effects of AgRP activation in subsequent figures.

RESPONSE. We thank the reviewer for the positive view on the revised manuscript.

Unfortunately, blood glucose data for the glucose gavage experiments (Figure 2) cannot be provided. The authors show that blood glucose levels are not appreciably different with AgRP activation in Figure 4c, but this was not in response to a glucose challenge. Measurements of glucose handling after a glucose challenge, including how quickly it is cleared, are different from measuring basal glucose blood levels. Perhaps a brief statement of this limitation could be included in the text. Technical reasons are cited in the response to reviewers.

RESPONSE. We agree with the reviewer and we have now included a statement on the limitation of baseline glucose levels.

The linear regression data between activity and RER is interesting, and it would certainly be anticipated that AgRP neuron activation in the absence of food would stimulate foraging behaviour. The wording on page 4 that describes the regression analysis could benefit from some re-wording (for example, what is meant by: (i) “anticipation in the effects of AgRP neuron activation”? (ii) “slopes been significantly higher than zero”?).

RESPONSE. We agree with the reviewer and we have edited the paragraph following the suggestions.

Figure 4 shows data to address the promotion of lipogenesis as an effect of AgRP activation. It can be noted that the difference in NEFA data (~ 1.6 vs 1.4 mEq/L) is quite small, even if statistically significant. To further implicate the promotion of lipogenesis as an effect of AgRP activation, the authors have now added in addition to their lipogenic data and C75 FAS inhibitor experiments a chronic

study over 10 days to show that AgRP activation increases fat mass to an extent not explained by increased food intake alone. Additionally, new loss of AgRP function experiments show that under positive energy conditions in mice fed through a gastric tube, AgRP neuron ablation leads to decreased fat mass gain. Was PNPLA2 gene expression measured? In WAT, key mechanisms regulating lipolysis involve not only HSL but also ATGL and perilipin (the latter particularly if beta-adrenergic stimulation is involved).

RESPONSE. We thank the reviewer for raising this important point. We have now measured PNPLA2 gene expression levels and reported in Figure 7. We did not observe any statistically significant changes in this gene upon ablation of Agrp neurons.

In the AgRP ablation study, during positive energy balance, is it known to what level AgRP neurons in control mice become activated? This question would be similar also control mice in the high-fat diet study, given that high-fat feeding also increases AgRP neuron activity. Do they ever reach levels comparable to AgRP-activated (intact) mice?

RESPONSE. This is a great question raised by the reviewer. No direct recordings of Agrp neurons in these metabolic conditions have been made. Our lab is trying to develop robust methods to infer neuronal activity in deep brain structures such as the hypothalamus during long recordings. This is, however, a non-trivial task as calcium reporters do not necessarily 'report' neuronal activity during these long recordings.

Of note, the authors have now also included exciting new experimental findings, including important experiments that begin to reveal the role of beta-adrenergic suppression, which mediates some of the metabolic effects of AgRP neuron activation (Figure 5). The use of the beta-adrenergic receptor agonist appears to completely reverse the metabolic effects of AgRP activation. Interestingly, the use of the beta-adrenergic receptor agonist, CL 316243, has previously been shown to decrease RER and increase plasma FFA in control and non-obese T2D mice (PMID: 16682489), which differs somewhat from the lack of effect that this agonist has in control mice.

RESPONSE. The reviewer raises the point of an apparent discrepancy between our results using CL 316243 and a previous report in the literature. This pharmacological agonist has substantial effects on lipolysis and food intake. However, these effects do not occur immediately after treatment and take several hours to begin. We have carefully considered this when designing our experiments. Purposively, we used acute injections of CL 316243 and metabolic measurements (no delay). In the manuscript cited by the reviewer, the metabolic measurements were collected during 3h with a delay of 1h after the treatment. These differences in experimental design could justify the lack of effect of CL 316243 on RER in our experiments.

Reviewer #2 (Remarks to the Author):

In this revision, the authors have addressed my points both experimentally and through further literature support. The new data have adequately addressed my concerns and recommend acceptance.

RESPONSE. We thank the reviewer for the positive view on the revised manuscript.

Reviewer #3 (Remarks to the Author):

The authors have addressed the Reviewers concerns in a satisfactory manner.

RESPONSE. We thank the reviewer for the positive view on the revised manuscript.

REVIEWERS' COMMENTS:

Reviewer #1 (Remarks to the Author):

This revised manuscript by Dr. Dietrich and colleagues includes requested revisions. The majority of my concerns are addressed in the manuscript and/or in the response-to-reviewers statements.